# De novo design of knotted tandem repeat proteins

Lindsey A. Doyle [1], Brittany Takushi[1], Ryan D. Kibler [2], Lukas F. Milles [2], Carolina T. Orozco[3], Jonathan D. Jones [3], Sophie E. Jackson [3], Barry L. Stoddard [1] ✉ & Philip Bradley [1,4] ✉

De novo protein design methods can create proteins with folds not yet seen in nature. These methods largely focus on optimizing the compatibility between the designed sequence and the intended conformation, without explicit consideration of protein folding pathways. Deeply knotted proteins, whose topologies may introduce substantial barriers to folding, thus represent an interesting test case for protein design. Here we report our attempts to design proteins with trefoil ($3_1$) and pentafoil ($5_1$) knotted topologies. We extended previously described algorithms for tandem repeat protein design in order to construct deeply knotted backbones and matching designed repeat sequences ($N = 3$ repeats for the trefoil and $N = 5$ for the pentafoil). We confirmed the intended conformation for the trefoil design by X ray crystallography, and we report here on this protein's structure, stability, and folding behaviour. The pentafoil design misfolded into an asymmetric structure (despite a 5-fold symmetric sequence); two of the four repeat-repeat units matched the designed backbone while the other two diverged to form local contacts, leading to a trefoil rather than pentafoil knotted topology. Our results also provide insights into the folding of knotted proteins.

The goal of protein design is to produce amino acid sequences that fold into well-behaved proteins and display desired structural and/or functional properties[1–4]. To date, most protein design approaches have relied on the structures of naturally occurring proteins as templates on which the designed sequences and structures are based. While these approaches have been highly successful in generating biomedically and biotechnologically useful protein constructs[5–8], they are limited to the subspace of protein structures nearby naturally evolved templates.

De novo protein design can create designed sequences and corresponding target folds that do not rely directly on naturally occurring (and previously visualized) structures, thereby sampling folds that are unlike those seen in nature to date[9,10]. In principle, arbitrarily complex topologies can be sampled by de novo protein design approaches; in

practice, however, de novo designs have tended to incorporate ideal structural features such as regular secondary structure elements connected by short, canonical linkers, strong hydrophobic patterning, and relatively simple folded topologies[11].

To further challenge our ability to generate more complex folded protein topologies through protein design, we have pursued the de novo design and subsequent characterization of two deeply knotted protein folds corresponding to the trefoil ($3_1$) and pentafoil ($5_1$) knots. Knotted proteins have been the subject of theoretical and experimental investigation since the early 1990s, when the possibility of such a topology was first examined and largely dismissed, albeit with the observation that one such structure (of human carbonic anhydrase) displayed a shallow knot corresponding to several terminal residues

¹Division of Basic Sciences, Fred Hutchinson Cancer Center, 1100 Fairview Ave. North, Seattle, WA 98109, USA. ²Department of Biochemistry, University of Washington, Seattle, WA 98195, USA. ³Yusuf Hamied Department of Chemistry, University of Cambridge, Lensfield Road, Cambridge CB2 1EW, UK. ⁴Division of Public Health Sciences and Program in Computational Biology, Fred Hutchinson Cancer Center, 1100 Fairview Ave. N, Seattle, WA 98009, USA. ✉e-mail: bstoddar@fredhutch.org; pbradley@fredhutch.org

being passed through a wide surface loop[12]. Since then, a growing collection of protein structures with deeper knots have been reported, culminating with a recent estimate that approximately 6% of deposited proteins are entangled via true knots or other topologies where the protein backbone passes through a closed loop[13,14]. The deepest knotted protein characterized to date (a functionally unannotated protein TP0642 from *Treponema pallidum*, PDB 5JIR) positions an entire folded protein domain on each side of a central protein knot[15]. The evolutionary pathway(s) leading to the formation of knotted proteins, as well as their folding pathways, biological purposes, and/or advantages are still not well understood, and are topics of continued investigation[16,17]. Because knotted proteins (which are found in all biological kingdoms) represent a small fraction of known protein structures and are highly diverse, few if any conserved features of sequence or structure exist that might inform de novo design of such features into a folded protein chain.

The computationally designed knotted proteins described in this study are related to previously described, de novo designed circular tandem repeat proteins (cTRPs), which have been constructed from repeated two-helix bundles and display a wide range of sizes and symmetries[9,18,19] (Fig. 1, left for an example). Those constructs represent a computationally designed subset of a much larger collection of naturally occurring and artificial tandem repeat proteins, all of which contain modular units of repeated protein sequence and structure, that can be composed of a variety of structural motifs including α-helical bundles, β-sheets, or mixed topologies[20–22]. They are particularly amenable to de novo design via purely computational approaches, due to their highly modular architectures[23,24]. To date, no helical tandem repeat proteins have been observed in nature that display entangled or knotted topologies. Therefore, we reasoned that extending our recent studies of designed cTRPs to incorporate such an uncommon structural feature and topology would represent a significant design challenge.

## Results

### Knotted protein design strategy

We developed an approach for designing tandem repeat proteins that fold into a class of knotted topologies known as torus knots. Torus knots can be embedded in the surface of an unknotted torus and are parameterized by a pair of integers $(p,q)$; the $(p,q)$ torus knot wraps $p$ times around the torus axis of rotational symmetry and $q$ times around the interior of the torus (Fig. 1). The approach builds on our previous work designing toroidal helical repeat architectures in which the N- and C-termini are juxtaposed. In that earlier work, we constructed circular N-repeat structures by constraining the repeat:repeat rotation angle to 360°/N, so that after N repeats the chain has completed one full turn about the central axis and the termini are juxtaposed. Recognizing that these structures could be viewed as simple $(p = 1, q = N)$ torus knots in which each of the repeats winds once about the interior of the torus (Fig. 1, left), we hypothesized that we could access genuine knotted topologies such as a $(p,q) = (2,3)$ trefoil or a $(2,5)$ pentafoil by doubling the angular rotation constraint (from 360°/N to 720°/N per repeat). More generally, a $(p,q)$ torus knot design could be achieved with a $q$-repeat protein whose inter-repeat rotation angle equals p * 360° /q. (Note that $p$ and $q$ must be relatively prime in order for the resulting structure to be comprised of a single protein chain). This simple modification to the backbone generation procedure produced compact, protein-like tandem repeat design models with the desired trefoil and pentafoil topologies (Fig. 1, center and right).

### Knotted trefoil designs and characterization

Four trefoil-knotted proteins (kcTRP3a, b, c, and d; 'knotted circular Tandem Repeat Proteins' with 3 repeats, variants a-d) were initially designed, corresponding to three moderately diverged architectures of repeating pairs of α-helices separated by short turns, folded into left-handed deeply knotted topologies (Fig. 1, center; sequences, alignments and additional illustrations of all constructs are provided in

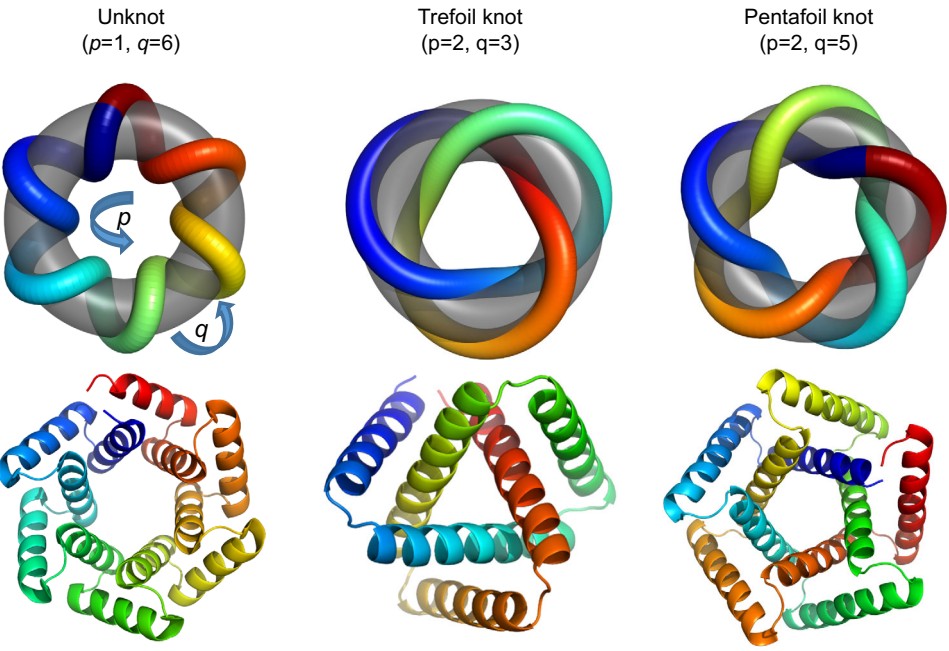

**Fig. 1 | Knotted tandem repeat proteins as torus knots.** Alpha-helical tandem repeat protein models (bottom row) considered as torus knots: curves embedded in the surface of the two-dimensional torus (top row). Torus knots are parameterized by a pair of integers $(p,q)$; the $(p,q)$ torus knot wraps $p$ times around the torus axis of rotational symmetry and $q$ times around the interior of the torus.

Models correspond (left to right) to an unknotted 6-repeat protein, a $(p = 2, q = 3)$ trefoil-knotted 3-repeat protein, and a $(p = 2, q = 5)$ pentafoil-knotted 5-repeat protein. Ribbon diagrams are colored as a rainbow spectrum, with the N-terminal helix blue and the C-terminal helix red.

**Table 1 | Crystallographic data collection and refinement statistics**

| PDB ID | kcTRP3d.1 7SQ3 | kcTRP3d.2 7SQ4 | kcTRP3d.3 7SQ5 | kcTRP5b.1 8ETQ |
|---|---|---|---|---|
| Data Collection | | | | |
| Space group | P 43 (No. 78) | P 21 3 (No. 198) | I 2 2 2 (No. 23) | P 43 21 2 (No. 96) |
| Unit cell | | | | |
| a, b, c | 57.2, 57.2, 56.0 | 55.81, 55.81, 55.81 | 47.233, 88.18, 95.257 | 59.6, 59.6, 178.2 |
| alpha, beta, gamma | 90, 90, 90 | 90, 90, 90 | 90, 90, 90 | 90, 90, 90 |
| Wavelength (Å) | 0.977 | 0.977 | 0.97741 | 0.97918 |
| Resolution range (Å) | 40.45–2.45 (2.54–2.45) | 39.46–1.49 (1.52–1.49) | 47.63–2.20 (2.24–2.20) | 44.55–2.42 (2.46–2.42) |
| Unique reflections | 6579 | 9669 | 10375 | 12655 |
| R-merge | 0.067 (0.391) | 0.040 (0.295) | 0.123 (0.705) | 0.146 (0.534) |
| R-meas | 0.070 (0.410) | 0.041 (0.303) | 0.129 (0.746) | 0.150 (0.559) |
| R-pim | 0.019 (0.121) | 0.007 (0.065) | 0.036 (0.239) | 0.031 (0.158) |
| CC1/2 | (0.947) | (0.983) | (0.814) | (0.984) |
| I/sigma(I) | 39.44 (4.0) | 103.8 (12.0) | 23.0 (3.0) | 20.2 (4.2) |
| Chi^2 | 0.983 (0.936) | 0.984 (0.993) | 1.062 (0.999) | 1.023 (0.930) |
| Multiplicity | 12.6 (10.5) | 35.8 (20.2) | 12.5 (9.4) | 21.2 (10.4) |
| Completeness (%) | 98.2 (85.0) | 99.2 (84.0) | 100 (99.6) | 99.9 (99.8) |
| Wilson B-factor | 55.51 | 19.39 | 29.88 | 29.9 |
| Refinement | | | | |
| R-work | 0.227 | 0.2274 | 0.2144 | 0.2023 |
| R-free | 0.2672 | 0.2656 | 0.2599 | 0.2545 |
| Number of non-hydrogen atoms | 1081 | 440 | 1118 | 2038 |
| Macromolecules | 1069 | 389 | 1069 | 1992 |
| Ligands | 2 | 17 | 2 | 0 |
| Water | 10 | 34 | 47 | 46 |
| Protein residues | 147 | 51 | 151 | 277 |
| RMS(bonds) | 0.006 | 0.0046 | 0.0074 | 0.005 |
| RMS(angles) | 0.8 | 0.69 | 0.78 | 0.64 |
| Ramachandran favored (%) | 97.92 | 100 | 100 | 98.89 |
| Ramachandran allowed (%) | 1.39 | 0 | 0 | 1.11 |
| Ramachandran outliers (%) | 0.69 | 0 | 0 | 0 |
| Clashscore | 7.67 | 2.3 | 4.22 | 4.63 |
| Average B-factor | 71.1 | 28.6 | 34.9 | 41.6 |
| Macromolecules | 71.1 | 26.9 | 34.6 | 41.59 |
| Ligands | 90.2 | 36.2 | 47.6 | N/A |
| Solvent | 68.3 | 45 | 39.6 | 41.9 |

Supplementary Table 1). In these designs, the N- and C-termini are in close proximity, yielding an almost perfect three-fold rotational symmetry. The designs kcTRP3a and 3d both containing a repeating unit corresponding to a 20-residue and 25-residue alpha-helix connected by a three-residue loop; kcTRP3b and kcTRP3c follow a similar architecture but contain repeating units corresponding to paired helices of 17 and 26 residues or 22 and 26 residues, respectively. The molecular weights (in kilodaltons) of the designed proteins are 18.7 (kcTRP3a), 17.2 (kcTRP3b), 20.0 (kcTRP3c), and 18.2 (kcTRP3d).

Of the four designs, three expressed readily in *E. coli* (kcTRP3b-d) while the remaining construct (kcTRP3a) displayed little to no expression (Supplementary Fig. 2). Of the three constructs that displayed visible expression, kcTRP3b was produced at the lowest levels and was not pursued further. kcTRP3c and kcTRP3d were both easily purified with multi-milligram yields per liter of bacterial culture. The two protein constructs eluted from the final SEC column in a single peak with a column retention volume corresponding to a protein monomer. Circular dichroism thermal denaturation analyses indicated no significant protein unfolding at temperatures up to 95 °C (Supplementary Fig. 3). Purified samples of kcTRP3c and kcTRP3d were easily

concentrated to protein concentrations of greater than 40 mg mL⁻¹. Subsequent crystallization screening experiments produced no diffraction-quality crystals.

The observation that kcTRP3c and 3d were well-behaved in solution, but did not readily crystallize, led us to conduct additional design computations, starting with those two models, with the intention of altering their stereochemical properties at symmetry-related, surface-exposed positions distributed along helix #1 in each repeat. The revised designs were intended both to alter the potential surface entropy and/or the stereochemistry at individual positions (by replacing a cluster of glutamate and/or lysine residues with alternative less polar and/or less flexible side chains) and to incorporate elements into the computations, that might further promote crystal growth via designed intermolecular contacts. A series of five redesigned models and constructs (two of kcTRP3c, named kcTRP3c.1 and 3 c.2, and three of kcTRP3d (kcTRP3d.1, 2, and 3) were generated (sequences, alignments, and illustrations provided in Supplementary Table 1). Except for kcTRP3c.1, which expressed at low levels, the newly created 'second generation' constructs behaved similarly to the original designs

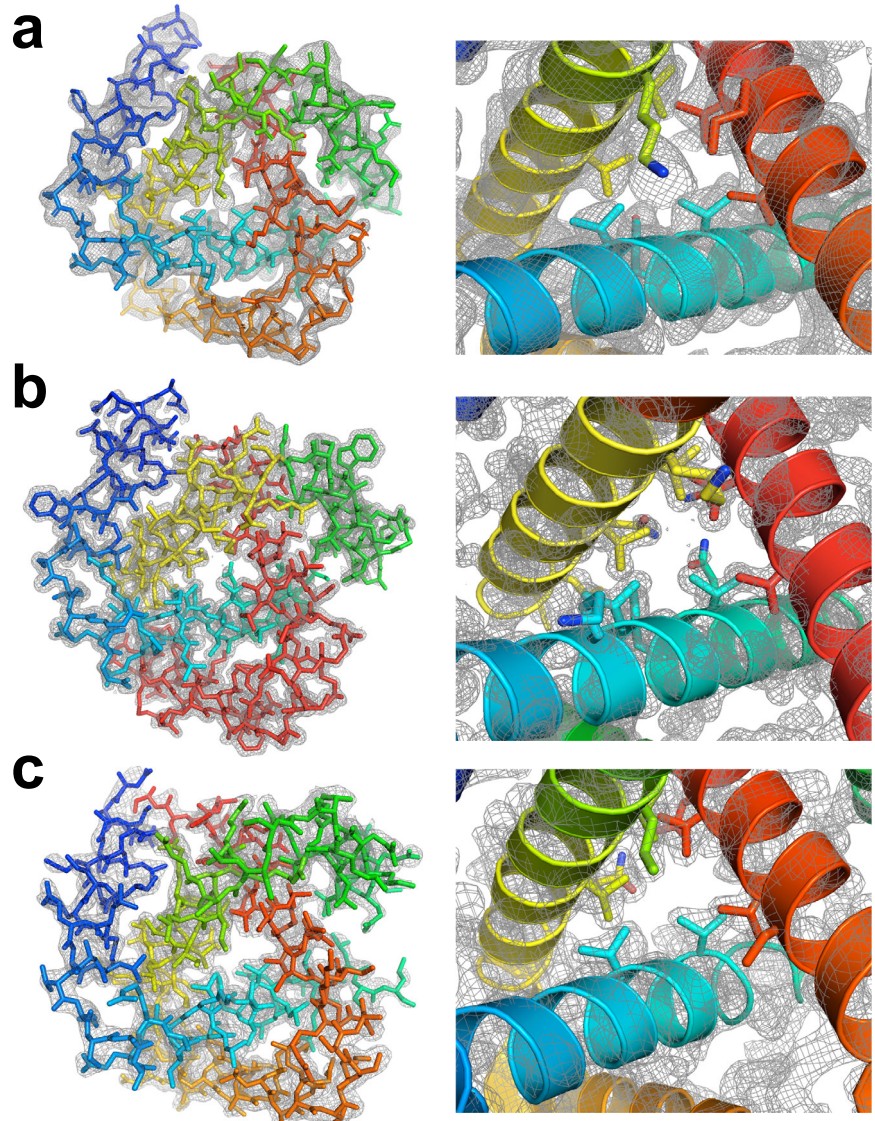

**Fig. 2 | Crystallographic structures of three related kcTRP3 trefoil designs.**
**a** Refined structure and 2Fo-Fc electron density map for kcTRP3d.1. Maps are displayed at 2σ contour level. **b** kcTRP3d.2. **c** kcTRP3d.3. See also Table 1 for crystallographic data and refinement statistics and Supplementary Table 1 and Supplementary Fig. 4 for additional details for all three constructs. Rainbow coloring of the molecular models is the same as in Fig. 1 (blue = N-terminal helix, red = C-terminal helix).

(Supplementary Fig. 4). While kcTRP3c.1 and 3c.2 remained recalcitrant to crystallization, crystals of kcTRP3d.1, 3d.2, and 3d.3 grew readily, and their structures were solved via molecular replacement. kcTRP3d.1 and 3d.3 were phased with the original designed atomic coordinates of kcTRP3d as a molecular probe. Initial molecular replacement runs of kcTRP3d.2 also used kcTRP3d as a probe, however these failed to produce a solution. Only when kcTRP3d was truncated to a single helix-turn-helix equivalent to a single repeating domain (or motif, i.e. one-third of the trefoil) did we obtain a solution, which contained a single repeating subunit sitting on the 3-fold crystallographic axis. When the Matthews coefficient is calculated with the number of molecules per asymmetric unit equal to a single repeat, the result falls within the expected range for globular macromolecules.

The crystal structures of kcTRP3d.1–3 (Fig. 2) validated the accuracies of the designed molecular models and their intended deeply knotted topologies. Superposition of the structures to the original designed atomic coordinates of kcTRP3d produced all-atom rmsd values of 1.84 Å (kcTRP3d.1), 1.43 Å (kcTRP3d.2), and 1.90 Å

(kcTRP3d.3). The structure of kcTRP3d.3 displays 120° rotational averaging in the crystal lattice, such that the positions between the repeating subunits contain a mixture of both peptide loops and protein termini. However, the conformations of secondary structure are unambiguous in electron density. This rotational averaging has been observed in previous studies on highly symmetrical repeat proteins[9]. The kcTRP3d.2 construct crystallized in the intended space group (P2₁3), with lattice dimensions and arrangement of molecules in the crystallographic unit cell that were similar to the lattice design model (55.35 Å vs 55.81 Å), but with differences in the lattice contacts corresponding to a register shift of one helical turn (Fig. 3). Since kcTRP3d.2 sits upon the 3-fold crystallographic axis, only a single repeating motif is present in the asymmetric unit (ASU). To obtain the full trefoil, crystallographic symmetry must be applied however this process averages the three repeating motifs of the trefoil into a single representation as well as averages out any remaining density from the flexible N-terminal affinity tag artifacts. This pseudo-symmetry and averaging are likely the cause of the higher-than-average R-values expected for a structure at this resolution.

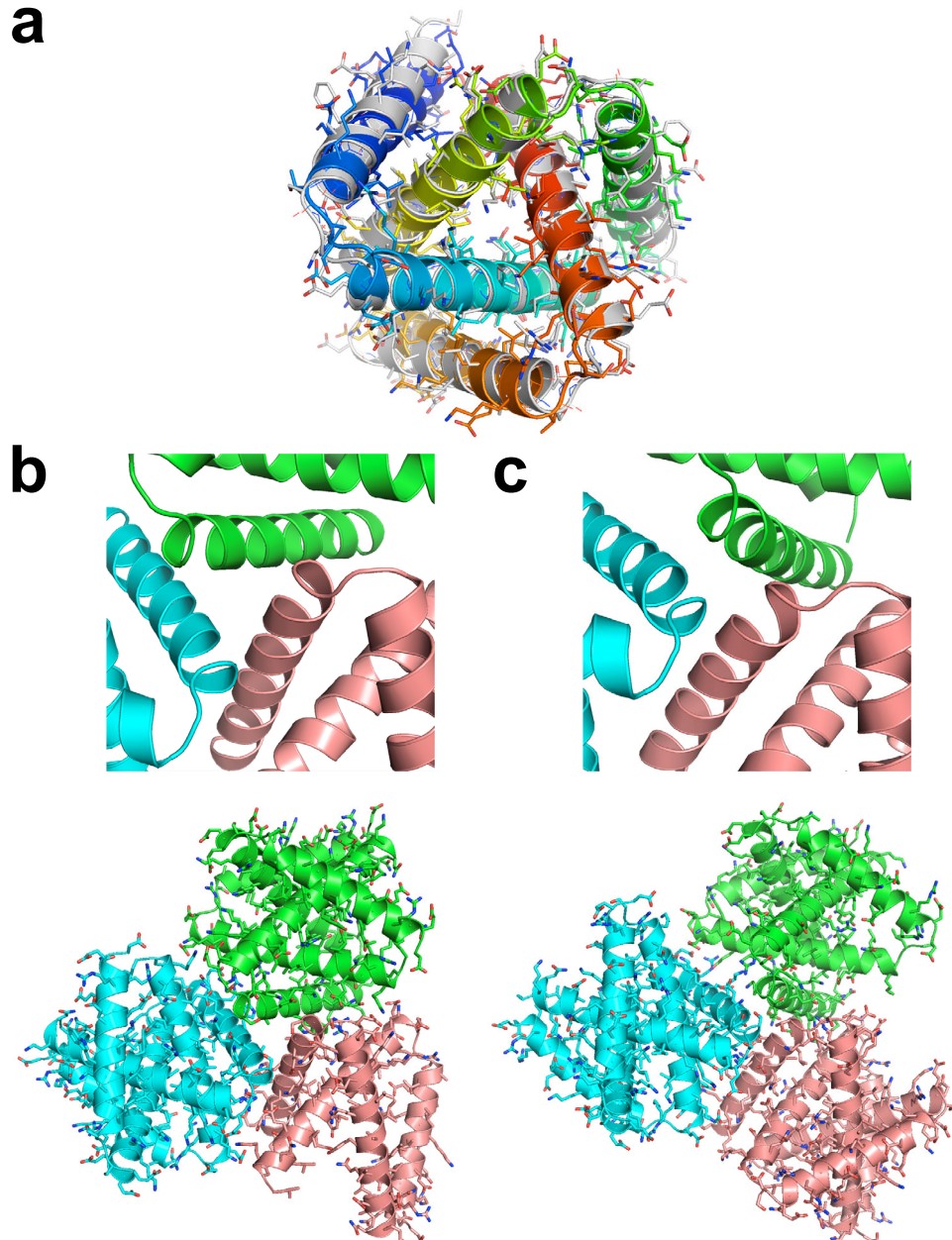

**Fig. 3 | Comparison of designed versus crystallographically observed protein folds and lattice contacts for kcTRP3d.2. a** Superposition of the designed construct (again colored as a spectrum; N-terminal helix is blue; C-terminal helix is red) and the corresponding crystal structure (colored gray). **b** Designed lattice contacts and packing between three protein subunits in the crystal lattice. **c** Experimentally observed lattice contacts between the three same protein subunits.

## Single-molecule unfolding studies

Atomic force microscopy (AFM) based single-molecule force spectroscopy (SMFS) was performed on kcTRP3d.1 to assess its single-molecule unfolding behavior. The designs were force loaded between their N- and C-terminus, using a specific covalent pulldown via the ybbr tag, and the ultrastable ClfA:Fgg peptide interaction (>2000 pN in rupture force) as a pulling handle (Fig. 4a).

Resulting force-distance traces acquired in constant velocity mode showed the expected sawtooth pattern with a single uninterrupted unfolding peak appearing at approximately 50–90 pN (Fig. 4b). Rupture force histograms showed the expected Bell-Evans distribution of rupture forces for a single ensemble[25] of unfolded contour length increments with an approximate Gaussian distribution, with the majority of measured lengths between 40 and 50 nm (Fig. 4e). These values are significantly shorter than the predicted length of a similar length construct in the absence of an internal knot (~60 nm, corresponding to 170 residues × 0.36 nm per residue).

The breadth of the length histogram and the shorter average length of individual molecules after rupture are consistent with the presence of a knot displaying somewhat variable 'tightness' upon application of force and rupture of surrounding tertiary structure. After the knotted proteins are unfolded, the remaining knot should tighten in the unfolded polypeptide chain. Potentially this could have effects on the polymer elasticity, or even premature hydrolysis of the backbone amide bonds at the high forces reached by the ClfA pulling handle. However, no such deviations were observed compared to non-knotted protein controls.

The tensile strength of this construct is considerably higher than other known purely alpha-helical domains such as spectrin (around 40–50 pN)[25]. However, the measured unfolding force is weaker than

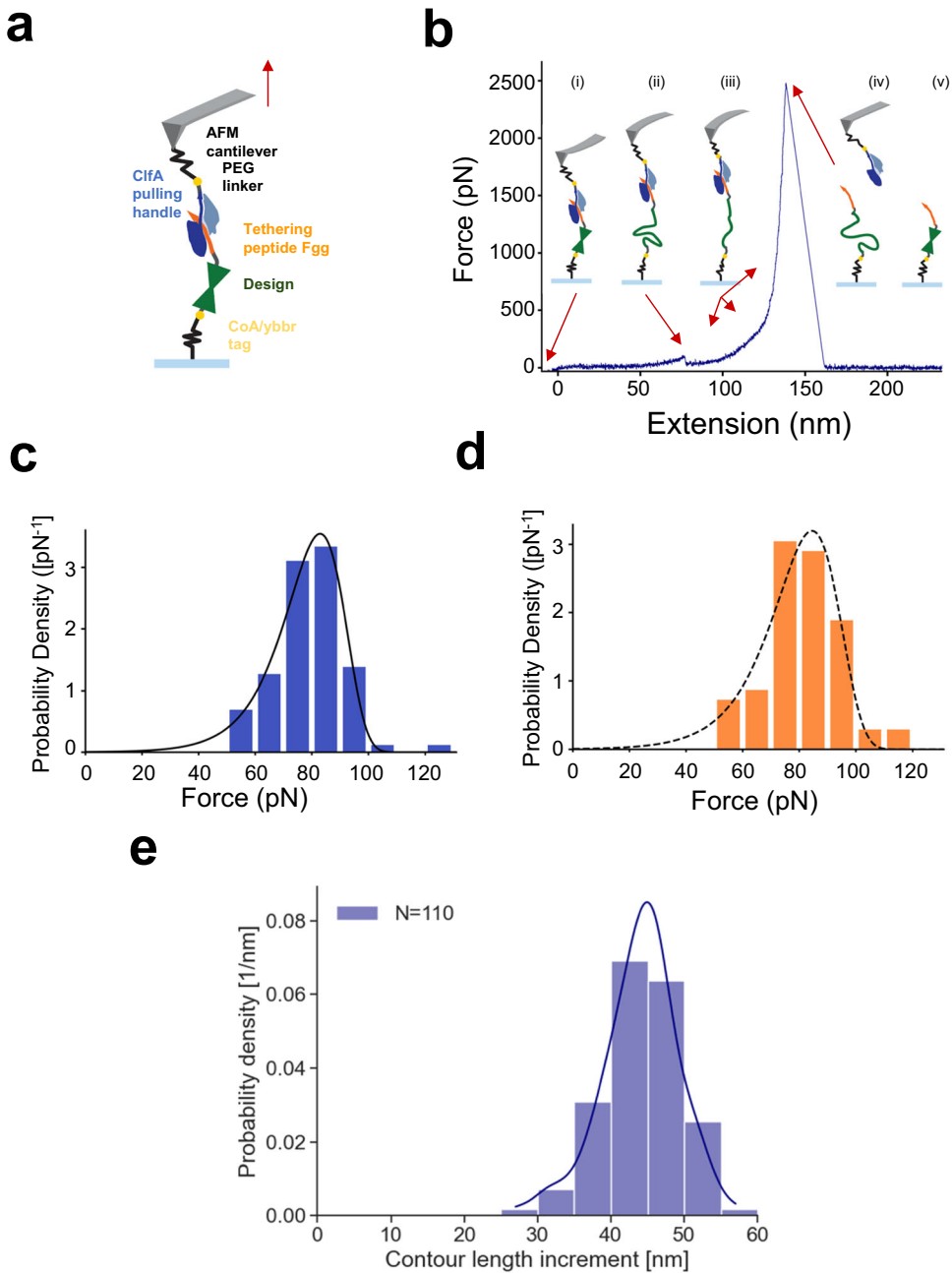

**Fig. 4 | Atomic force microscopy analysis of the physical unfolding properties of kcTRP3d.1. a** Experimental setup for AFM-SMFS: Proteins are covalently anchored to surface and cantilever by polyethylene glycol (PEG) via the ybbr-tag (yellow). Force is applied to the design using the Fgg (orange) peptide on its C-terminus by its binding partner ClfA (blue) immobilized on the cantilever. **b** Representative force distance curve for unfolding of designs. (i) The AFM cantilever is pushed into the surface allowing the ClfA-Fgg interaction to form. The cantilever is then retracted at a constant velocity of 1.6 μm/s until the design unfolds resulting in a sharp force drop around 80 pN indicated by the red arrow (ii). Now the full length of the unfolded polypeptide chain of the design is stretched (iii), until finally the ClfA:Fgg interaction ruptures at around 2500 pN (iv). The design then could potentially refold as the mechanical stress is now relieved (v). **c** Unfolding force spectrum for the crystallized variant, assembled from $n = 67$

individual unfolding events, including a fit of the Bell-Evans model (solid line, Δx = 0.40 nm, koff0 = 2.5E–1 1/s, most probable rupture force 83 pN). **d** Similar unfolding force spectrum for the original design sequence, $n = 69$ individual unfolding events, including a fit of the Bell-Evans model (dashed line, Δx = 0.36 nm, koff0 = 3.6E–1 1/s, most probable rupture force 85 pN). The unfolding force spectra in **c** and **d** cannot be distinguished with a 2-sided KS test ($p = 0.64$), indicating that the underlying stability of these constructs is identical within resolution of the assay. **e** Histogram of the unfolded, freed contour lengths measured for design unfolding and corresponding Kernel Density estimate (line) with a most probable contour length around 45 nm, which is shorter than the expected contour length increment of 60 nm for complete unfolding of a 170 residue protein. Source data are provided as a Source Data file.

other naturally occurring proteins such as muscle protein Titin IG27 (around 250 pN)[26], as well as other de novo designed proteins such as Top7 (approximately 150 pN)[27], or the ultrastable pulling handle ClfA (around 2400 pN)[28] at similar force loading rates. Those constructs derive their strength from sheared, paired beta strands. It is interesting

to speculate whether the tensile strength of kcTRP3d.1 that was determined here is related to the presence of the knot, or some other aspect of the design process such as optimized non-covalent interactions. Although some computational studies have suggested that a knot can increase the mechanical force required to unfold a protein

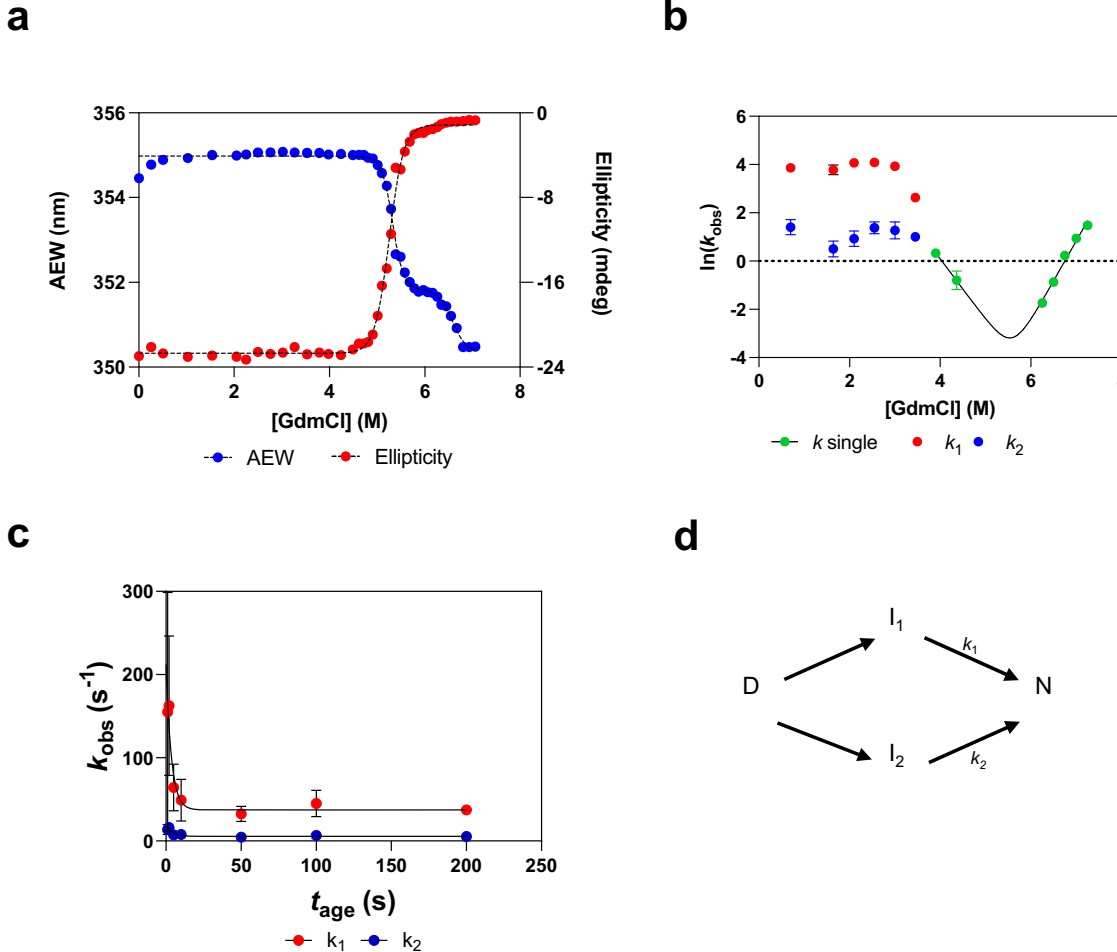

**Fig. 5 | Thermodynamic and kinetic stability of kcTRP3d.3 at pH 7, 25 °C.**
**a** GdmCl- induced unfolding of kcTRP3d.3 under equilibrium conditions probed by intrinsic fluorescence, plotted as the average emissions wavelength (AEW) (solid blue circles) or by far-UV CD, plotted ellipticity at 222 nm (solid red circles). The dashed black lines show the best fit of the data to a three- or two-state unfolding model, for the fluorescence and far-UV CD data, respectively. **b** Unfolding and refolding rate constants determined at different GdmCl concentrations. Unfolding kinetics were measured between 3.5 and 6.75 M GdmCl and data fitted to either a single or double exponential function to obtain unfolding rate constant(s) at each GdmCl studied (solid green circles). Data represented as mean values ± SD, $n = 3$. Refolding kinetics were measured by first fully denaturing kcTRP3d.3 in 7.2 M GdmCl to ensure refolding was from the denatured state. Below 3 M GdmCl, refolding kinetics best fit to a double exponential function which generated two

refolding rate constants, the faster one, termed $k_1$, is shown in solid red circles, whilst the slower phase, termed $k_2$, is shown in solid blue circles. Data represented as mean values ± SD, $n = 3$. **c** Interrupted unfolding experiments showing the refolding rate constants as a function of ageing time. Data from the faster phase are shown as solid red circles, and data corresponding to the slower phase are represented by solid blue circles. The solid line shows the best fit of the data to a single-exponential function which generates rate constants 0.2–0.3 s$^{-1}$. Data are presented as mean +/− SD, where $n = 3$. **d** Schematic for the refolding of kcTRP3d.3 from its denatured state (D), along parallel pathways on which an intermediate $I_1$ or $I_2$ is populated. The rate-determining step in refolding is assumed to be from $I_1$ or $I_2$ to N and the refolding from D to $I_1$ or $I_2$ is assumed to be sufficiently fast that it has occurred within the deadtime of the stopped-flow instrument.

structure[29], experimental results have shown that for all the knotted proteins studied to date, the force required for unfolding is well within the range found for unknotted protein structures[30]. This is also the case whether the protein structure is mechanically unfolded to a knotted or unknotted state[31]. As the tensile strength of other unknotted designed proteins is also high[27], it is thought likely that the high tensile strength of kcTRP3d.1 studied here is due to the optimization processes and not the knot.

## Thermodynamic stability of the trefoil-knotted protein

The thermodynamic stabilities of kcTRP3d.1, kcTRP3d.2, and kcTRP3d.3 were studied under equilibrium conditions using the chemical denaturant guanidinium chloride (GdmCl) and unfolding was monitored using both the intrinsic fluorescence and far-UV CD signal of the protein to probe both tertiary and secondary structure. Figure 5a shows the results obtained at pH 7.5 and 25 °C. Two unfolding transitions were observed with midpoints at 5.3 and 6.6 M GdmCl in

the fluorescence experiments, whilst a single unfolding transition was observed by far-UV CD, with a midpoint of 5.2 M GdmCl. For kcTRP3d.2 and kcTRP3d.3, unfolding in GdmCl was shown to reach equilibrium after 24 h of incubation at 25 °C (Supplementary Fig. 5a, b), and for kcTRP3d.3, duplicate measurements were performed at pH 7.5, 25 °C to establish reproducibility (Supplementary Fig. 5c). In addition, unfolding experiments were also undertaken for kcTRP3d.3 at 10 °C, and kcTRP3d.2 and kcTRP3d.1 at pH 7.5 at 25 °C (Supplementary Fig. 5c–e). All data were fit to either a two- or three-state unfolding model. All three constructs show similar stabilities and the stability of kcTRP3d.3 did not change significantly between 25 and 10 °C. Collectively, these data establish that all three constructs unfold and populate an intermediate state, in which a considerable amount, if not all, of the α-helical structure has been lost but in which there remains some burial of the tryptophan side chain(s) suggesting that hydrophobic interactions play a role in stabilizing the intermediate state. In all cases, the free energy change between intermediate and native state are over

20 kcal mol⁻¹, showing that the native states of the proteins are all highly stable.

The stability of kcTRP3d.3 was also measured at pH 4 (Supplementary Fig. 5f). The results show that the protein remains very stable at this lower pH value. Interestingly, the intermediate state retains some α-helical structure under these conditions as shown by the second unfolding transition observed in the far-UV CD data (Supplementary Fig. 5f).

**Unfolding and refolding kinetics of the trefoil-knotted protein**
Both unfolding and refolding kinetics of kcTRP3d.3 were studied using stopped-flow kinetics and GdmCl. In the unfolding experiments, native kcTRP3d.3 was rapidly mixed with solutions containing high concentrations of GdmCl to induce unfolding, which was monitored using intrinsic fluorescence. Kinetic data were fit to a single exponential to obtain the rate constant for unfolding, $k_u$, at a range of final GdmCl concentrations, and the natural logarithm of $k_u$ was plotted against the concentration of GdmCl (Fig. 5b). These data can be used to estimate the half-life of unfolding in the absence of chemical denaturant, which is ~100 years; again, illustrating how stable this protein is. The fact that a single exponential is observed despite the population of an intermediate state shown under equilibrium conditions, is consistent with the fact that over the range of GdmCl concentrations studied, unfolding is to the intermediate state only.

The refolding kinetics of kcTRP3d.3 were also studied by first fully denaturing the protein in 7.2 M GdmCl, before rapidly diluting it into low concentrations of chemical denaturant in the stopped flow. The refolding kinetics were measured over a range of GdmCl concentrations from approx. 1 to 5.5 M. From 3 to 5.5 M final GdmCl, the kinetic traces fit well to a single exponential function to generate a single refolding rate constant, $k_F$. At lower GdmCl concentrations, from 1 to 3 M, the kinetic data fit better to a double exponential to generate two refolding rate constants, $k_F^1$ and $k_F^2$. The natural logarithm of these rate constants is plotted against final GdmCl (Fig. 5b). Whilst the values of $\ln(k_F)$ increase with decreasing GdmCl concentration from 5.5 to 3 M as is usual, the values below 3 M roll over such that the folding rate becomes relatively independent of denaturant concentration for both of the refolding phases observed. These results suggest that either (i) an intermediate state is populated under these conditions, (ii) the protein may be transiently aggregating, (iii) the denatured state is heterogenous with a slow process leading to the formation of a slower-folding population of molecules, or (iv) the process intrinsically does not depend considerably on GdmCl concentration indicating that it may well not be associated with a large change in structure.

We systematically investigated the first three reasons for rollover. To eliminate the possibility that the rollover is due to the population of the intermediate state observed in the equilibrium experiments, the experiment was repeated but the initial unfolded protein was in 6.2 M GdmCl. In this case, refolding from the stable intermediate not the denatured state is probed. The results were very similar to those obtained in 7.2 M, suggesting that the rollover is not due to the stable intermediate observed in previous experiments. However, it may still result from additional intermediates only observed under kinetic conditions. To eliminate the possibility that rollover is due to transient aggregation, the experiments were repeated at higher protein concentrations up to 5 μM, and results were the same as obtained at 1 μM, indicating that aggregation is not an issue (Supplementary Fig. 5g, h). To further analyze the two refolding phases observed at lower GdmCl concentrations, the relative amplitudes were calculated and plotted against denaturant concentration. For most concentrations, the fastest refolding phase corresponds to about 75% of the overall amplitude whilst the slower of the two refolding phases constitutes about 25% (Supplementary Fig. 5i,j). If the two refolding phases are due to parallel pathways from the denatured to the native state, then one would expect kinetic partitioning with relative amplitudes in line with the

ratio of the rate constants of the two phases, this would be 90% and 10%. As such, it can be assumed that parallel pathways directly from D to N do not occur.

To eliminate the possibility that the two refolding phases are due to heterogeneity in the denatured state, e.g, to proline isomerization (or, as there are no prolines in kcTRP3d.3, due to some other isomerization event), interrupted unfolding experiments were performed. In this case, native protein was rapidly unfolded in high concentration of GdmCl, aged for some time, $t_{age}$, and then rapidly mixed with refolding solutions and the refolding kinetics measured at different ageing times. Generally, in these experiments, one expects the rate constants to be independent of ageing time but for the amplitudes of the refolding phases to vary. We observed an unusual result, in that the refolding rate constants decreased with ageing time from 212 s to 37 s⁻¹, and 18 to 5.5 s⁻¹ (Fig. 5c). The change in refolding rate constants with ageing time was fitted to a single exponential decay and the rate constant for both phases was found to be 0.2–0.3 s⁻¹, suggesting both phases are affected by the same process. As it has been shown that knotting generally slows down folding, we hypothesize that the process observed in the denatured state may be unknotting of the polypeptide chain following unfolding of secondary and tertiary structure.

**Designed pentafoil proteins**
Motivated by our success in designing and validating the structure of deeply knotted trefoil proteins, we turned to a more complex knotted topology: a 'pentafoil' or $5_1$ knot (Fig. 1, right), which has not yet been observed in the protein structure database. Three initial pentafoil constructs were generated, corresponding to two additional moderately diverging architectures that again contained repeating pairs of α-helices separated by short turns folded into left-handed deeply knotted topologies. Design kcTRP5a contains a repeating unit corresponding to 24-residue and 22-residue alpha-helices connected by three-residue loops. Designs kcTRP5b and kcTRP5c follow a similar architecture except the leading helix is extended by four residues, to a total of 28. Characterization and validation of the initial pentafoil designs (Supplementary Fig. 6) proved to be more challenging than for the trefoil designs, with significant precipitation and sample loss upon concentration. Crystallization trials were unsuccessful, and small angle x-ray scattering (SAXS) analysis of kcTRP5a suggested higher than expected Rg and evidence of flexibility (Supplementary Fig. 7).

Given the complexity of the designed pentafoil topology, and inspired by the substantial advances in machine learning methods for protein sequence design and model validation that had occurred since the original pentafoil designs were generated, we attempted a second round of design. We retained the kcTRP5a-b backbones while redesigning their sequences using the fixed-backbone design algorithm ProteinMPNN[32], enforcing sequence repeat symmetry. We then filtered the ProteinMPNN-designed sequences to ones that the AlphaFold2 network[33] predicted were likely to fold into the intended topology. Despite the lack of natural sequence homologs, we and others have found that AlphaFold2 can generate highly accurate models for successful de novo designs while also flagging design failures[34]. Indeed, retrospective AlphaFold2 modeling of the trefoil designs yielded predictions that agreed with the design models for 3 of 4 constructs, whereas the first generation pentafoil sequences were predicted to fold into unknotted helical barrels (Supplementary Fig. 8).

Of six second-generation pentafoil constructs that were ultimately examined in the laboratory (three each based on kcTRP5a and kcTRP5b/c), five were successfully cloned into bacterial expression vectors, of which three were successfully expressed at high levels and purified to homogeneity as described in methods (Supplementary Fig. 9). One construct, based on the kcTRP5b backbone and referred to here as kcTRP5b.1, yielded crystals that diffracted to 2.4 Å resolution. A molecular replacement search with the kcTRP5b.1 model yielded a

highly significant solution; subsequent refinement and iterative model building revealed good agreement between the design and crystal structure across the core alpha helices (Fig. 6a), but divergence in two of the four repeat-repeat loop connections (Fig. 6b and Supplementary Fig. 10a) leading to a globally dissimilar topology. Despite the perfect repeat symmetry in sequence, the observed structure is asymmetric (Fig. 6c, d), with two repeat connections (R1-R2 and R4-R5) matching the design and two diverging to enable local repeat-repeat packing (R2-R3 and R3-R4). As a result of these topological differences, the observed structure forms a trefoil knot rather than the intended pentafoil knot (Supplementary Fig. 10b). All local and non-local residue interactions formed in the design model are also observed somewhere in the crystal structure (allowing for equivalence between repeats); thus, it is tempting to speculate that the designed sequence is in fact compatible with the intended pentafoil topology, but folds to a more kinetically accessible trefoil topology under experimental conditions.

## Discussion

### Protein design and knotted topologies

Within the structural biology and protein folding community, the long-held belief was that deeply knotted protein structures would not exist, the entanglement of a polypeptide chain into a knot being incompatible with postulated protein folding mechanisms[12,35,36]. However, it has been established now for several decades that even deeply knotted protein structures exist[37], and that proteins can adopt a number of knotted topologies including trefoil ($3_1$), figure-of-eight ($4_1$), three-twist ($5_2$), and stevedore ($6_1$) knots[13,38,39]. Moreover, two very recent studies have used the results of AlphaFold2 predictions and potentially identified more complex knotted protein structures, including $5_1$, $6_3$, and $7_1$ knots, as well as composite $3_1-3_1$ knots[40,41]. The $5_1$, $6_3$, and $7_1$ knotted structures have yet to be verified experimentally, however, a $3_1-3_1$ knotted protein structure has recently been solved demonstrating this topology does occur naturally[42]. Collectively, these studies suggest that a wider range of knotted protein structures may exist than originally thought, including structures that would require more complex folding pathways with two, not just one, threading events.

Even though examples of knotted protein structures have been known for some time, there have been very few attempts to rationally design a knotted structure. The Yeates group took inspiration from the internal pseudo symmetry within the known trefoil-knotted protein (VirC2) that comprises two ribbon-helix-helix binding domains similar to the Arc repressor, suggesting that it may have evolved from gene duplication and fusion, to design a trefoil knotted protein based on the homodimeric structure of the *Helicobacter pylori* protein HP0242[43]. In that case, the C-terminus of one of the monomers in the dimer lies close in space to the N-terminus of the other monomer, and engineering of a short linker sequence to connect the two was successful in generating a monomeric left-handed trefoil knotted structure. Although this was an excellent example of how to use an evolutionary approach to engineer a knotted protein structure, it did not develop any methods for designing knotted structures de novo. There is one other example of a designed protein that contains a knot (PDB 3CWO)[44], but in that case the aim was not to introduce a knot, and the knot itself is very shallow.

Our design results have a few broader implications. On the one hand, the successful design of a trefoil-knotted protein using an approach that focuses exclusively on native-state stabilization shows that explicit consideration of folding pathways or misfolded conformations is not strictly required, even for the design of knotted topologies. In agreement with the results of the Yeates group[43], we find that a protein chain can fold to its knotted native state without having been subjected to evolutionary pressure for that property. On the other hand, the failure of the $5_1$ pentafoil design which, unlike the naturally occurring $5_2$ structure already observed, requires a more complex folding pathway in which there are two, not one, threading events, suggests that designing such knotted structures may be far more challenging. In this respect, it will be interesting to see whether the $5_1$ and $6_3$ knotted protein structures recently predicted[40,41] by AlphaFold2 are verified experimentally. Our results suggest that whilst the $5_1$-structure may be thermodynamically stable, it may be much less kinetically accessible than singly-threaded topologies. The only pentafoil design that could be structurally characterized at high resolution, kcTRP5b.1, misfolded to an alternate topology with significantly lower contact order than the design model. This alternate structure showed striking agreement with the design model in the packing of the inner and outer alpha helices (Fig. 6a), but the sequence order of the helices in the barrel differed due to changes in the conformation of two of the four inter-repeat connections. This misfolding may have been promoted by the exact sequence identity between the five tandem repeat units, which enabled one repeat in the observed structure to take the place of another in the design model while conserving the amino acid identity of nearly all the pairwise residue interactions. Given that the observed structure validates essentially all the local and non-local interactions in the design model (allowing for equivalence between repeats), it may be possible to rescue the design by introducing changes to the sequence that break repeat symmetry and destabilize the misfolded structure.

### Similar topologies in naturally evolved (and previously engineered) proteins

A comprehensive analysis of the structure, stability, mechanical unfolding and knot tightening, under force, unfolding and refolding kinetics of the designed trefoil knotted proteins described here, enables a comparative analysis of the structure and properties of these proteins with naturally occurring (and several previously engineered constructs, described above) left- and right-handed trefoil-knotted proteins in the KnotProt database[38].

Within the database, there are 537 examples of protein chains that have been shown to adopt a + $3_1$ right-handed trefoil knot, although there is some doubt about 9 of these due to chain breaks in the structure[38]. Two families (carbonic anhydrase and α/β knotted methyl transferases) account for more than 84% of the chains; in total there are nine distinct protein structures that adopt this knotted topology: carbonic anhydrase, α/β knotted methyltransferase, methionine adenosyl transferase (MAT), transcarbamylase, $Ca^{2+}/Na^+$ (or $H^+$) exchanger, ribosome assembly protein, NAIP5, TP00624 (a *Treponema pallidum* protein of unknown function) and a very shallow-knotted protein created by a protein design project where fragments of two other protein folds were fused to form a chimeric sequence (for which the authors did not set out to make a knotted protein). These proteins vary in size, structure and both the depth of the knot (here, shallow knots are defined as those where nine or less residues can be truncated from either N- or C-termini before the knot is lost and deep over 10 residues), and also the size of the knotted core, i.e, the number of residues that form a loop through which the chain must pass to form the knot. Most, but not all, of the right-handed trefoil knotted proteins are enzymes.

In contrast, the database contains only 32 examples of protein chains that form a −$3_1$ left-handed knot (in four examples there is some uncertainty about the knotted topology because of chain breaks). However, there are nearly as many distinct folds in these 32 examples as there are for the right-handed trefoil knots. Eight different folds form this type of knot, including proteins in the spliceosome, DNA phosphothioation, mitochondrial ribosome, DNA-binding protein (virC2), RNA-binding, pre-ribosome assembly product, and a designed knotted protein based on the structure of HP0242 from *H. pyloris*. Whereas some of these are enzymes, many more in this category have other functions. As with the right-handed trefoil knotted proteins, they display a wide range of knot depths and core sizes.

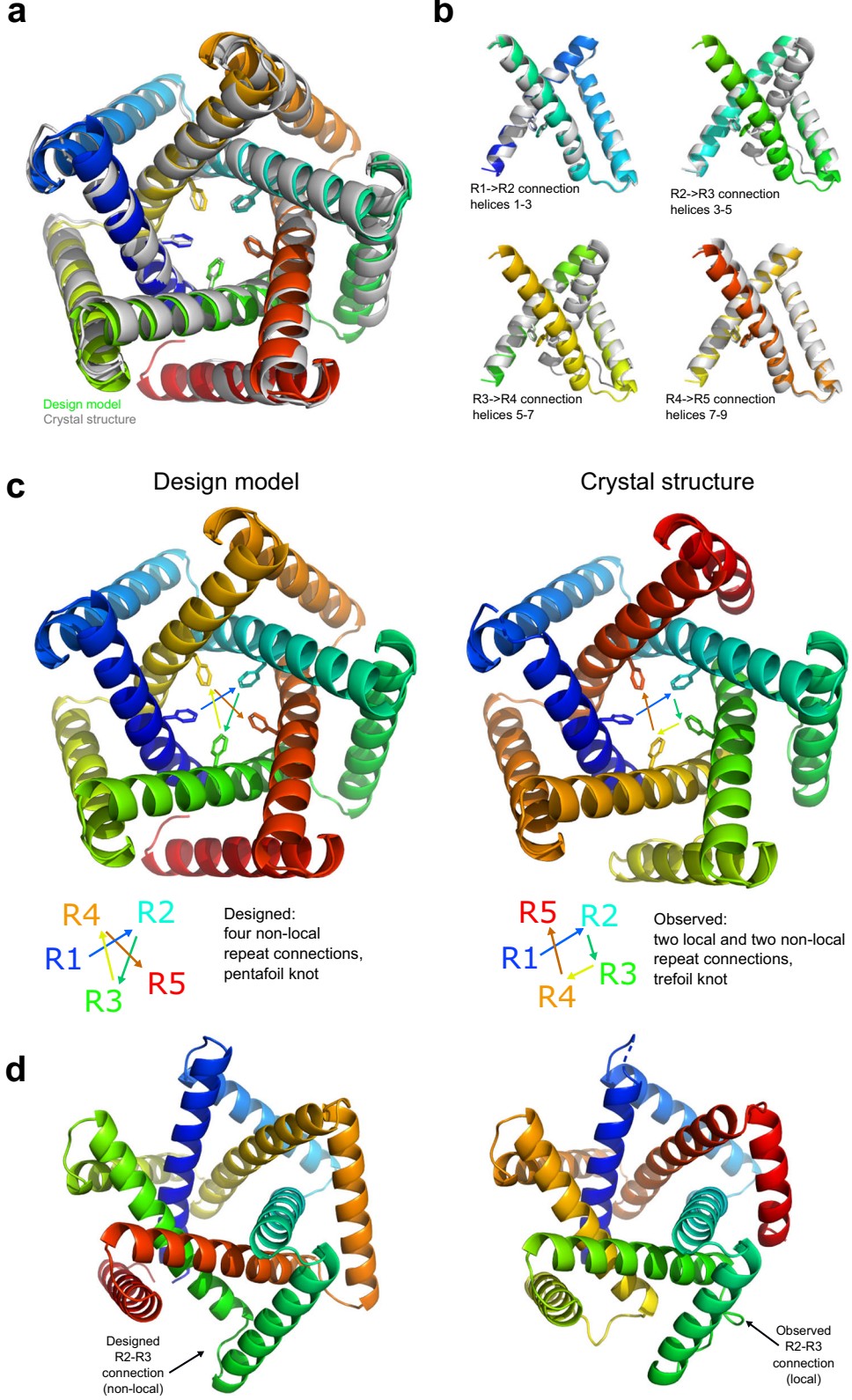

**Fig. 6 | Comparison of the designed and observed structures for pentafoil design kcTRP5b.1. a** Superposition of the design model (colored by sequence number, from blue at the N-terminus to red at the C-terminus) and the crystal structure (gray) shows good agreement in the placement of the alpha helices. **b** Unlike the design, and despite its perfect 5-fold sequence symmetry, the observed structure is asymmetric, with two of the four repeat-repeat connections showing good agreement with the design model while the remaining two diverge substantially. **c** Divergent topologies can be seen in the side-by-side view of the design model and crystal structure, with the design making 4 non-local repeat-repeat connections, while the observed asymmetric fold has two local and two non-local connections. The observed fold remains knotted, but forms the topologically simpler trefoil knot rather than the intended pentafoil knot. **d** Tilted view showing the impact of divergence in the repeat-repeat loop connections.

The computationally designed trefoil-knotted proteins described here form a left-handed trefoil-knotted structure that is distinct from any of the left- or right-handed knotted proteins known to date and discussed above. They form a moderately deep knot and are significantly more stable both thermodynamically and kinetically than any trefoil knotted protein that has been characterized so far[43,45–47]. Similar to the deeply knotted MTases YibK and YbeA, and the designed knotted protein 2ouf-knot, it adopts a partially folded intermediate state under equilibrium conditions, in contrast to the shallow-knotted MJ0366[47]. Such intermediate states have also been observed for many other unknotted proteins of similar length and stability, illustrating that this result is not necessarily a direct result of the knot. However, it has considerably less secondary structure in the intermediate state than any of these other examples. We speculate that this, with the fact that the native and intermediate state show unusual fluorescence properties, may result from tryptophan residues which are, unusually, on the surface of the protein in the native state, but at least partially buried in the intermediate structure. This suggests that the intermediate state is, in part, stabilized by burial of hydrophobic surface area.

Given that it has been suggested that a knotted topology may confer stability on a protein[13], it is interesting to speculate whether the superstability reported here for kcTRP3d.3 comes from the knot or some other aspect of the design such as optimized sequences for the formation of helices and/or optimized packing in the core. Given that other de novo design projects have created proteins with stabilities that are above the typical range measured for naturally occurring sequences[9,10], and that experimentally it remains unclear whether knots confer stability on proteins or not[16], it seems more likely that it results from the optimal interactions introduced as part of the design process. The superstability of kcTRP3d.3 is also reflected in the half-life of unfolding in water, which is estimated to be some 100 years – far greater than measured for any other knotted protein[16]. It is interesting to note that despite the considerable thermodynamic stability measured with chemical denaturants, the mechanical stability of kcTRP3d.3 measured by AFM is similar to those observed for other knotted proteins, as well as within the range of forces measured for unknotted proteins[16]. However, the tightened knot in kcTRP3d.3 is somewhat larger than those observed previously for either a $4_1$-knotted phytochrome[48], or a $3_1$-knotted unfolded state of UCH-L1, and in fact is closer in size to the $5_2$-knotted unfolded state of UCH-L1[31]. It is likely that this is because of the exact location of the knot in the unfolded state and the size of the side chains within the tightened knot.

## Refolding behaviors of natural and designed knotted proteins

As with many knotted proteins, apart from MJ0366 which is both small and has a very shallow knot, the refolding kinetics of kcTRP3d.3 are complex. There are multiple refolding phases observed at low but not higher concentrations of GdmCl, and their dependence on denaturant concentration changes significantly at about 3.3 M GdmCl. It is easier to consider first the refolding phase observed between 3.3 and 5 M GdmCl. This refolding data along with the unfolding data can be fit to a simple two-state model (Supplementary Fig. 11) and compared to the parameters obtained for the transition between intermediate (I) and native (N) states measured under equilibrium conditions. In this case, the unfolding and refolding kinetic data are within error of the thermodynamic parameters, indicating that this kinetic unfolding and refolding phase corresponds to the unfolding and refolding of native to intermediate state and vice versa. The two refolding phases observed at lower concentrations of chemical denaturant are more challenging to interpret as such kinetics are consistent with a number of different mechanisms, three of which are described in Supplementary Fig. 12. All three schemes are similar in that they have two parallel pathways from the denatured to the native state, an intermediate on at least one pathway and a change in the rate-determining step for folding. The evidence in support of each pathway is described in Supplementary Fig. 12. We favor a mechanism in which refolding from the denatured state can occur along two pathways on both of which there is a metastable intermediate state (Fig. 5d and Supplementary Fig. 12- Scheme III). Such a mechanism has previously been proposed for another knotted protein, UCH-L1, that contains a $5_2$-knot. This knot is, however, relatively shallow compared to some of the very deeply knotted proteins, and only a single intermediate is observed under equilibrium conditions[49]. For UCH-L1 and kcTRP3d.3, it is likely that the second intermediate is also present under such conditions but is either spectroscopically similar to the first intermediate with comparable energetics, and/or is present only in small amounts and therefore difficult to observe in ensemble fluorescence or far-UV CD measurements.

In contrast to UCH-L1, for kcTRP3d.3 we also have evidence from the interrupted unfolding experiments that the denatured state initially populated on unfolding (which we term D*), undergoes a conformational change over time (with a half-life of ~200−350 ms), which we hypothesize is associated with untying of the knot to the denatured state observed in equilibrium conditions (D). The two fast refolding phases observed at GdmCl concentrations below 3.3 M, correspond to the protein folding to $I_1$ and $I_2$. The very low dependence of these phases on denaturant concentration suggests a process in which there is rather little change in solvent accessible surface area between the ground state (D) and the transition state. We speculate that these may be associated with hydrophobic collapse and a threading step in a state that does not have significant structure. At 3.3 M, there is a switch in the rate-determining step from D → I to I → N. The refolding kinetics associated with the I → N step, are consistent with the results from the stability measurements which indicate that this step is associated with the formation of much of the helical secondary structure. This explains, in part, the very fast kinetics of this phase when extrapolated to water, (the rate constant in water is estimated to be $8 \times 10^5 \, s^{-1}$) comparable to some measurements on the rate of formation of an α-helix[50]. Of importance is the fact that, although the folding of many knotted proteins has often been reported[16] or predicted[13] to be slow, kcTRP3d.3 folds rapidly in water with a similar rate to that observed for much smaller proteins with simple topologies such as CI2[36].

## Methods

### Constructs and nomenclature

The construct names, sequences, and relationships between all constructs described in this paper are provided and illustrated in Supplementary Table 1. The constructs described in this article are referred to as kcTRPs ('knotted circular Tandem Repeat Proteins'). Following the nomenclature used for previous cTRP constructs[9,18], designed constructs are annotated as kcTRP3 and kcTRP5, where the designed proteins contain a total of 3 or 5 repeats, respectively. An additional annotation at the end of those names distinguish between individual original designed 'siblings' (for example, four different unique designs of kcTRP3 constructs were initially created and tested, and are designated as kcTRP3a, kcTRP3b, etc.). Subsequent 'second generation' designs that were created from a common parent are further denoted with an additional numerical annotation (for example, a second round of designs that were all created from the parental 'kcTRP3d' design are designated as kcTRP3d.1, kcTRP3d.2 and kcTRP3d.3).

### Computational design

The designed trefoil knot proteins reported here are tandem repeat proteins with three or five identical sequence repeats. The N- and C-termini are juxtaposed, yielding a closed structure with near-perfect 3-fold or 5-fold rotational symmetry (Fig. 1). The sequence was designed by modifying a method we previously developed[9,18,19] for

design of closed tandem repeat proteins in which the rotation and translation between repeat units is constrained during backbone assembly and refinement in order to give closed structures. In this method, the transformation between successive repeats is factored into a rotation about an axis and a translation parallel to that axis. To achieve a closed structure, the magnitude of the rotation is constrained to equal 360°/N (where N is the number of repeats), and the translation is constrained to equal 0. Recognizing that our previous unknotted designs could be viewed as torus knots (curves embedded in the surface of a simple torus) in which each repeat wraps once around the torus and the protein chain passes once around the central symmetry axis (Fig. 1, left) we modified the constraint on inter-repeat geometry to produce models that passed twice around the central symmetry axis. In other words, rather than constraining the inter-repeat rotation to 360°/N, we tethered it to 720°/N. For three-repeat designs, this yielded a trefoil knot topology ($3_1$ knot or (2,3) torus knot; Fig. 1, center), and for 5-repeat designs it produced models with a pentafoil topology ($5_1$ knot or (2,5) torus knot, a topology that has not yet been observed in the protein databank; Fig. 1, right).

To promote crystallization of an initial round of trefoil designs that expressed and purified well, we subsequently performed additional lattice design using Rosetta's lattice modeling framework[51], aligning the 3-fold symmetry axis of the designs with 3-fold axes of the target crystal lattice, randomly sampling the remaining orientational degrees of freedom, and decreasing the lattice dimensions until lattice contacts were formed. A large number of candidate designs were generated in different space groups and with different orientational parameters and ranked based on the predicted interface energy between monomers, filtering for unsatisfied buried hydrogen bond donors and acceptors.

A second round of five-fold symmetric pentafoil sequence designs were generated by applying the ProteinMPNN fixed-backbone protein design network algorithm[32] to the backbone models from the first round, enforcing sequence repeat symmetry and using either 0.1 Å or 0.3 Å of backbone noise during inference with weights trained with no backbone noise (epoch51_step255000 parameter set). We filtered the ProteinMPNN-designed sequences to ones that the AlphaFold2 network[33] predicted were likely to fold into the intended topology, as assessed by a mean pLDDT score greater than 80. Filtered designs were ranked by spatial aggregation propensity (SAP) score[52] and the top 3 sequences for each of the kcTRP5a and kcTRP5b backbones were chosen for experimental characterization.

## Protein expression and purification

All coding sequences used in this study (Supplementary Table 1) were commercially synthesized and ligated between NcoI and NotI restriction sites in an in-house T7 bacterial plasmid expression vector, pET15HE[53] (GenScript Inc.) and illustrated in Supplementary Fig. 1. In the resulting constructs, the protein coding sequences were preceded by an N-terminal 6-histidine purification tag and a thrombin cleavage site for removal of the affinity tag. Plasmids were transformed into BL21-CodonPlus(DE3)-RIL *Escherichia coli* cells (Agilent Technologies, 230245) and plated on LB medium with 100 µg mL⁻¹ ampicillin. Expression of designs were done either by autoinduction[54] or addition of Isopropyl β-d-1-thiogalactopyranoside (IPTG). Briefly, a 10 mL LB medium culture containing 100 µg mL⁻¹ ampicillin was inoculated with a single transformant and grown 16–20 h at 37 °C with shaking at 220 rpm. The entire 10 mL culture was added to 1 L of LB medium containing 100 µg mL⁻¹ ampicillin and grown at 37 °C with shaking at 220 rpm until the culture reach log phase with an optical density at 600 nm of between 0.6 and 0.8. IPTG was added to a final concentration of 0.5 mM and the culture was grown at 16 °C for 16–20 h with shaking at 220 rpm. Cultures containing expressed protein were pelleted by centrifugation and stored at −20 °C.

Frozen cell pellets were thawed at room temperature and resuspended in 50 mL of 1× phosphate-buffered saline ('PBS'; 137 mM NaCl, 10 mM $Na_2HPO_4$, 2.7 mM KCl, pH 7.4.) and phenylmethylsulfonyl fluoride (PMSF) was added to a final concentration of 0.5 mM. Cells were lysed via sonication at 70% power with four cycles of 20 s on and 60 s off. Lysate was cleared of cell debris by centrifugation in a SS34 rotor at 30,597×g for 20 min at 4 °C followed by manual filtration through a 5 micron filter. Clarified lysate was added to 2 mL of nickel-NTA metal affinity resin (Invitrogen, R90115) equilibrated with PBS and incubated at 4 °C for one hour with agitation. The mixture was applied to a gravity-fed column, the resin washed twice with PBS plus 25 mM imidazole and protein eluted by three applications of 5 mL PBS plus 300 mM Imidazole. Elutions containing the protein were pooled, concentrated and buffer exchanged into PBS containing no imidazole. Elutions were split in half and the poly-Histidine tag was removed from one half via Thrombin digest with -0.2 U of biotinylated thrombin protease added per 1 mg of protein and incubation at 25 °C for 16–20 h with gentle rocking. The protease was then removed by incubation with Streptavidin resin. The tagged and/or tag-free protein were then applied to a 0.2 mm filter and run over a size exclusion column (Cytiva HiLoad 16/600 Superdex 75 pg, 28989333, or Superdex 200 pg, 28989335, or Supderdex 200 10/300 GL, 17517501) equilibrated in 25 mM Tris at pH 7.5 plus 200 mM NaCl. Appropriate fractions were pooled and concentrated.

## Circular dichroism

Purified, tagged proteins were dialyzed into 10 mM potassium phosphate buffer at pH 7.0 for 16–20 h at 4 °C and diluted to approximately 25 micromolar protien concentration as determined by using the molecular weight and extinction coefficient on a NanoDrop One spectrophotometer (Thermo Fisher). Diluted samples were filtered through a 0.2 µ filter and wavelength scans from 190–250 nm were performed at 25 °C and 95 °C (JASCO J-815 spectrometer with a Peltier temperature controller).

## Crystallization and structure determination

Various designed trefoil proteins (Supplementary Table 1), all with their N-terminal affinity tags proteolytically removed, were first screened for crystals using several broad matrix screening kits and a Mosquito drop-setting robot (TTP LabTech). Crystal hits were then optimized in a 24-well hanging drop tray in a 2 µL drop with a protein to well volume ratio of 1:1. The kcTRP3d.1–3 trefoils were crystallized in the following conditions: 2.0 M DL-Malic acid pH 7.0 (kcTRP3d.1 at 10 mg mL⁻¹), 0.1 M sodium acetate pH 4.5 and 35% 2-Methyl-2,4-pentanediol (kcTRP3d.2 at 11 mg mL⁻¹), and 0.2 M potassium sodium tartrate, 0.1 M sodium citrate pH 5.6 and 1.8 M ammonium sulfate (kcTRP3d.3 at 17 mg mL⁻¹). The kcTRP5b.1 pentafoil design, with its N-terminal affinity tag attached, was crystallized in 0.2 M sodium chloride, 0.1 M imidazole pH 8.0, 0.15 M sodium phosphate monobasic, and 1.35 M potassium phosphate dibasic.

Crystals were cryoprotected prior to plunging into liquid nitrogen by addition of 25% glycerol (kcTRP3d.1) or 20% sucrose (kcTRP3d.3 and kcTRP5b.1) to the well solution. Protein kcTRP3d.2 did not require addition of a cryo agent prior to flash freezing. Data were collected on beam line 5.0.1 at the Advanced Light Source in Berkeley, CA (kcTRP3d.1–3) or beam line 19-ID at the Advanced Proton Source in Lemont, IL (kcTRP5b.1) at 0.9762 Å x-ray wavelength and processed with HKL2000[55]. Phases were determined via molecular replacement with the original computational design in the Phenix suite[56] with Phaser[57], followed by rounds of refinement with Phenix refine and model building with Coot[58].

## Single molecule unfolding analyses via atomic force microscopy

Detailed AFM-SMFS protocol have been published previously[59,60]. In brief, AFM Cantilevers (Biolever Mini AC40TS, Olympus, Tokyo, Japan)

were modified with Aminosilane. Following 15 min of UV-Ozone cleaning (Novascan, USA), cantilevers were silanized via submersion in 1 mL (3-aminopropyl)-dimethyl-ethoxysilane (APDMES, Abcr Inc, Karlsruhe, Germany or Gelest Inc, USA) mixed with 1 mL ethanol and 5 μL ultrapure $H_2O$ for 5 min. Each cantilever was rinsed in ethanol and subsequently in ultrapure $H_2O$. Finally, cantilevers were baked at 60 °C for 1 h to be stored overnight under Argon and used in the following steps the next day. Aminosilanized glass surfaces (76×26 mm, 1 mm thickness) were purchased from Schott Glaeser (Nexterion A+) and kept under Argon until used. Both glass surfaces and cantilevers were covered with 5 kDa heterobifunctional α-maleinimidohexanoic-PEG-NHS (Rapp Polymere, Tübingen, Germany or Jenkem Technology, USA) dissolved in 75 mM HEPES (pH 7.5) at 25 mM (100 mg mL$^{-1}$) for 30 min. After rinsing surfaces and cantilevers in ultrapure water, 1 mM Coenzyme A (in 50 mM sodium phosphate pH 7.2, 50 mM NaCl, 10 mM EDTA buffer) was applied to both for at least 1 h. CoA functionalized surfaces and cantilevers stored in coupling buffer (50 mM sodium phosphate pH 7.2, 50 mM NaCl, 10 mM EDTA buffer) at 4 °C were stable for more than four weeks. Two protein constructs were compared, where the pulling handle was ClfA-ybbr – a high force adhesin from *S. aureus* that binds the Fg gamma peptide (Fgg). Designs were ordered as ybbr-DESIGN-Fg-gamma peptide. Where the ybbr tags served as covalent surface pulldowns and the ClfA:Fgg peptide was used to apply force to the C-terminus of the design.

When different protein constructs were compared with a single cantilever, up to 10 spatially separated spots were created using a silicone mask (CultureWell reusable gaskets, Grace Bio-Labs, Bend, OR, USA), cleaned in isopropanol and ultrapure $H_2O$, dried in a gentle stream of nitrogen, heated to 60 °C and securely pressed onto a silanized microscopy slide (Nexterion A+, Schott). Pegylation and CoA coupling in individual wells was achieved following identically to the protocol described above[61].

These steps yielded cantilevers and surfaces covalently coated in PEG-CoA. Cantilevers and surfaces were rinsed in ultrapure water. Protein functionalization was achieved by covalently pulling down proteins via their ybbr-tag to CoA by the SFP enzyme coupling. The proteins of interest were diluted into TBS150 (25 mM Tris, 150 mM NaCl, pH 7.4) supplemented with 10 mM $MgCl_2$. Cantilevers were typically incubated with 50 μM of protein, here ClfA-ybbr, and 3 μM Sfp phosphopantetheinyl transferase (SFP) for at least 1 h. The glass surfaces were incubated with 5 – 15 μM of protein construct of interest and 1–2 μM SFP for 30–60 min, depending on the desired surface density. Both samples were rinsed extensively with at least 50 mL measurement buffer (TBS75: 25 mM Tris, 75 mM NaCl, pH 7.4) buffer before experiments.

Data were acquired on a custom-built AFM operated in closed loop by a MFP3D controller (Asylum Research, Santa Barbara, CA, USA) programmed in Igor Pro 6 (Wavemetrics, OR, USA). Experiments were conducted at room temperature (~25 °C). Cantilevers were briefly (<200 ms) and gently (<200 pN) brought in contact with the functionalized surface and then retracted at constant velocity of 1.6 μm/s. After each curve acquired, the glass surface was moved horizontally by at least 100 nm to expose an unused, fresh surface spot. Typically, 50,000 curves were recorded per experiment. To calibrate cantilevers the Inverse Optical Cantilever Sensitivity (InvOLS) was determined as the linear slope of the most probable value of typically 40 hard (>2000 pN) indentation curves. Cantilevers spring constants were calculated using the equipartition theorem method with typical spring constants between 70–160 pN nm$^{-1}$ [62,63] A full list of calibrated spring constants from experiments presented in this work is provided below, as the stiffness of the pulling handle, i.e. the cantilever, may influence the complex rupture and domain unfolding forces measured.

Data analysis was carried out in Python 2.7 (Python Software Foundation)[64–66]. Laser spot drift on the cantilever relative to the calibration curve was corrected via the baseline noise (determined as the last 5 % of datapoints in each curve) for all curves and smoothed with a moving median (windowsize 300 curves). The inverse optical lever sensitivity (InvOLS) for each curve was corrected relative to the InvOLS value of the calibration curve.

Raw data were transformed from photodiode and piezo voltages into physical units with the cantilever calibration values: The piezo sensitivity, the InvOLS (scaled with the drift correction) and the cantilever spring constant (k). The last rupture peak of every curve was coarsely detected and the subsequent 10 nm of the baseline force signal were averaged and used to determine the curve baseline, that was then set to zero force. The origin of molecule extension was then set as the first and closest point to zero force. A correction for cantilever bending, to convert extension data in the position of the cantilever tip was applied. Bending was determined through the forces measured and was used on all extension datapoints (x) by correcting with their corresponding force datapoint (F) as xcorr = x - F/k.

To detect unfolding or unbinding peaks, data were denoised with Total Variation Denoising (TVD, raw, not denoised, data shown in plots)[67,68], and rupture events detected as significant drops in force relative to the baseline noise. A three-regime polymer elasticity model by Livadaru et al.[69]. was used to model the behavior of contour lengths freed by unfolding events and transformed into contour length space[70] (Livadaru model parameters were: stiff element $b = 0.11$ nm and bond angle $\gamma = 41°$). A quantum mechanical correction was used to account for peptide bond stretching at high forces[71]. Especially at forces larger than 1 nN this correction was essential to be able to fit the data to polymer elasticity models accurately. Peaks were assigned their contour length in diagrams assembled through Kernel Density Estimates (KDE) of the contour length transformed force-extension data. The KDE bandwidth was chosen as 1 nm. The loading rate was fitted as the linear slope of force vs. time of the last 4 nm preceding a peak.

For single Bell-Evans (BE) model at a given force loading rate r (determined as most probable loading rate from all unfolding events through a KDE) with the parameters Δx and $k_{off,0}$ the probability density p(F, r, Δx, $k_{off,0}$) to unfold at a given force F was fit to a normalized force histogram.

## Chemical denaturation analyses

Stock protein solutions were prepared in 50 mM Tris pH 5.5, at 60 μM protein concentration. Chemical denaturation curves corresponding to 41 protein samples were prepared with guanidinium chloride (GdmCl) ranging from 0 to 7.1 M final concentration: 220 μL of denaturant solution was mixed with 20 μL of protein to a final concentration of 5 μM. The buffer solutions were dispensed with a liquid handling robot (Microlab®500 Series, ML541C, Hamilton Company). The denaturant solutions mixed with the protein were incubated at 25 °C at different time points until they reached equilibrium (7 days). Each of the denaturation samples were analyzed in a 100 μL quartz cuvette for the intrinsic fluorescence measurements (Hellma, Precision Cell in Quartz SUPRASIL®, Typ No: 105.250-QS, Light Path: 10×2 mm, Center: 20 mm), and in a 350 μL quartz cuvette for the far UV-CD measurements (Hellma, Absorption Cell in Quartz Glass High Performance, Article No: 110-1-40, Optical path length: 1 mm, Center height: 8.5–20 mm). The fluorescence was recorded with a Cary 400 Eclipse Fluorescence Spectrophotometer (Agilent Technologies) set by thermostat at 25 °C and controlled by a heat block: the samples were excited at 280 nm, the emission was recorded from 300 to 400 nm, with a scan rate of 600 nm min$^{-1}$, excitation and emission band passes were set at 10 nm. The far UV-CD signal was measured on a Chirascan CD Spectrometer (Applied Photophysics), thermostatted at 25 °C controlled by a heat block: the UV-CD absorbance was measured at 222 nm with a bandwidth of 2 nm, for 30 s per sample.

The intrinsic fluorescence data were analyzed using an average emission wavelength (AEW), which is the arithmetic mean of the wavelengths weighted by the fluorescence intensity at each

wavelength. It is calculated as shown in Eq. 1:

$$AEW = \frac{\sum_{i=1}^{N} F_i . \lambda_i}{\sum_{i=1}^{N} F_i} \tag{1}$$

where $F_i$ is the intensity of fluorescence at each wavelength, and $\lambda_i$ is the wavelength.

If protein unfolding curves appeared two-state, a two-state model was used to fit the AEW data:

$$F = (\alpha_N + \beta_N.[den]) + \frac{(\alpha,|D + \beta_D.[den]).exp\left(\frac{m_{D-N}}{RT}.\left([den] - [den]_{50\%}\right)\right)}{1 + exp\left(\frac{m_{D-N}}{RT}.\left([den] - [den]_{50\%}\right)\right)} \tag{2}$$

When an intermediate between the native and the denatured was sufficiently stable to be populated and observed, a three-state model was used. The equilibrium between the different species is as follows:

$$N \rightleftharpoons I \rightleftharpoons D \atop K_{I-N} \; K_{D-I} \tag{3}$$

The denaturation curves, using the AEW data, were fitted to a three-state model using Eq. 4:

$$F = \alpha_N + \beta_N.[den] + \alpha_I.exp\left(\frac{m_{I-N}}{RT}.\left([den] - [den]_{50\% I-N}\right)\right) + \frac{(\alpha|D + \beta_D.[den]) exp\left(\frac{m_{I-N}}{RT}.\left([den] - [den]_{50\% I-N}\right)\right) exp\left(\frac{m_{D-I}}{RT}.\left([den] - [den]_{50\% D-I}\right)\right)}{1 + exp\left(\frac{m_{I-N}}{RT}.\left([den] - [den]_{50\% I-N}\right)\right) + exp\left(\frac{m_{I-N}}{RT}.\left([den] - [den]_{50\% I-N}\right)\right) exp\left(\frac{m_{D-I}}{RT}.\left([den] - [den]_{50\% D-I}\right)\right)} \tag{4}$$

where $\alpha_N, \alpha_I, \alpha_D$ are the fluorescence of the native, intermediate and denatured states in $H_2O$ respectively; $\beta_N, \beta_D$ are the slopes of native and denatured baselines respectively; $m_{I-N}, m_{D-I}$ are the $m$-values between the intermediate and native state, and denatured and intermediate states respectively; $\Delta G_{I-N}^{H2O}, \Delta G_{D-I}^{H2O}, \Delta G_{D-N}^{H2O}$ are the differences in Gibbs free energy between the intermediate and native states, denatured and intermediate states, and denatured and native states, respectively; T is the temperature and R is the gas constant.

## Unfolding and refolding kinetic analyses

For unfolding kinetic experiments, the native protein and the denaturant solutions were mixed in a 1:10 ratio, respectively. Six stock solutions of guanidinium chloride were prepared in 50 mM Tris pH 7.5 so that the final concentrations ranged from 6.25 to 7.25 M GdmCl with an interval of 0.25 M. The protein stock solution was prepared at 60 μM to achieve a final concentration of 5.5 μM after mixing with the denaturant solutions. For the refolding kinetics experiments, the was initially denatured at a high concentration of chemical denaturant (8 M GdmCl), to ensure it was totally unfolded before mixing with refolding solutions containing no or low concentrations of chemical denaturant. Eight stock refolding solutions of low concentrations of guanidinium chloride (GdmCl) were prepared in 50 mM Tris pH 7.5 such that the final GdmCl concentrations ranged from 0.7 to 4.4 M GdmCl final. The denatured protein stock solution was 60 μM.

The unfolding and refolding kinetics were monitored with a SX20 stopped-flow spectrometer from Applied Photophysics (software: SX Spectrometer Control Panel Application version 2.2.27). The temperature of the water bath was set to 25 °C, the excitation wavelength was set to 280 nm, both slit widths were 2 mm. A cut off filter of 320 nm was used. The unfolding and refolding curves were fitted with the software Pro DataViewer version 4.2.27. The fluorescence signal corresponding to the unfolding was fitted with a single-exponential function (Eq. 5) whilst the refolding kinetics were fit to a single or double-exponential function (Eq. 6).

$$A(t) = A_1.exp(-k_1 t) + c \tag{5}$$

$$A(t) = A_1.exp(-k_1 t) + A_2.exp(-k_2 t) + c \tag{6}$$

where $A_1$ and $A_2$ are the amplitudes, $k_1$ and $k_2$ the respective unfolding or refolding rate constants and $c$ the offset.

The natural logarithm of the unfolding rate constants was then calculated and plotted *versus* denaturant concentration, and the data fitted with Eq. 7.

$$lnk_U^{[den]} = lnk_U^{H_2O} + m_{k_U}[den] \tag{7}$$

Where $k_U^{[den]}$ is the observed unfolding rate constant at the denaturant concentration $[den]$, $k_U^{H_2O}$ is the unfolding rate constant in water and $m_{kU})$ is the slope of the plot of $lnk_U^{[den]}$ versus denaturant concentration. For the interrupted-unfolding, double-jump experiments, native protein in 50 mM Tris pH 7.5 was initially rapidly mixed in a 1:5 ratio with a solution of 8.3 M in 50 mM Tris pH to induce unfolding. This solution was aged for some time $t_{age}$, before it underwent a second mixing in a 1:5 ratio with a refolding solution containing 2.5 M GdmCl in 50 mM Tris pH 7.5. The final protein concentration was 1.25 μM and the final GdmCl concentration was 3.2 M. Data were fitted to a double-exponential function.

## Small angle x-ray scattering analyses

Small angle X-ray scattering data were collected at the SIBYLS beamline[72] at the Advanced Light Source in Berkeley, CA, USA. Before collection, samples were placed in a 96-well plate. Each sample was presented to the X-ray beam using an automated robotics platform. The 10.2 keV monochromatic X-rays at a flux of $10^{12}$ photons s$^{-1}$ struck the sample with a 1 × 0.3 mm rectangular profile that converged at the detector to a 100 μm × 100 μm spot. The detector-to-sample distance was 2 m and nearly centered on the detector. Each sample was exposed for a total of 10 s. The Pilatus 2 M detector framed the 10 s exposure in 300 ms frames for a total of 33 frames.

kcTRP5a with purification tags removed was purified by gel filtration (buffer corresponding to 0.1 M Tris/HCl pH 8.0, 0.5 M NaCl and 1 mM EDTA) and monomer peak fractions were pooled and concentrated using a Millipore Amicon Ultra-15 10 K molecular weight cutoff concentration filter. Buffer corresponding to flow-through was saved for use as a reference blank for buffer subtraction in the SAXS experiment. Scattering data were collected at SIBYLS beamline 12.3.1 at the Advanced Light Source. Data was collected at protein sample concentrations of 3 mg ml$^{-1}$ and 1.5 mg ml$^{-1}$ and for two blank samples. Buffer subtraction was performed by SIBYLS and resulted in six datasets (two buffer subtractions and an average of both buffer subtractions (three datasets) for each of two samples). There was no evidence of radiation damage in any of the datasets; therefore program Frameslice (https://sibyls.als.lbl.gov/ran; version 1.4.13) at the SIBYLS beamline[72] was used to average all 32 individual exposures for each datasets to produce averaged scattering profiles. The FoXS web server[73] was used to compare the design model to each averaged experimental scattering profiles and calculate quality of fit (X) values. The c2 parameter was fixed at 0 to avoid overfitting. The short

n-terminal peptide "MARS" and the c-terminal peptide remaining following Thrombin cleavage "GSLVPR" were not present in the protein models used for generating the theoretical spectra. The dataset with the best signal-to-noise and X value was selected for further analysis. Guinier analysis for determination of Rg and the dimensionless Kratky analysis were performed using ScÅtter (ScÅtterIV).

### Reporting summary

Further information on research design is available in the Nature Portfolio Reporting Summary linked to this article.

## Data availability

The structures described in this manuscript have been deposited in the RCSB protein database under PDB ID codes 7SQ3, 7SQ4, 7SQ5, and 8ETQ. Comparative analyses of natural knotted protein structures utilized the KnotProt database v2.0 (https://knotprot.cent.uw.edu.pl/). The original source data and raw images corresponding to biochemical and biophysical analyses have been uploaded to the Harvard Dataverse public repository (https://dataverse.harvard.edu/dataverse/kcTRPs/) and are also available upon request from the authors. Source data are provided with this paper.

## Code availability

The protein design simulations described here were conducted with the Rosetta software suite (www.rosettacommons.org). Further details on command line flags, input files, and protocol-specific source code for knotted protein design can be found in the software repository Github https://github.com/phbradley/knotted_designs.

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

## Acknowledgements

We thank Meredith Steward for assistance with biochemical experiments. Support for this work was provided by the Fred Hutchinson Cancer Center and by the National Institute for General Medical Sciences (NIGMS) for both BLS (R01 GM139752) and PB (R35 GM141457). L.F.M. was supported by a Human Frontier Science Program Cross Disciplinary Fellowship (LT000395/2020-C) and an EMBO Non-Stipendiary Fellowship (ALTF 1047-2019). Crystallographic data collection was conducted at the Advanced Light Source (ALS) at the Berkeley Center for Structural Biology which is supported in part by the Howard Hughes Medical Institute. The Advanced Light Source is a Department of Energy Office of Science User Facility under Contract No. DE-AC02-05CH11231. The Pilatus detector at Beamline 5.0.1 (where the data was collected) was funded under NIH grant S10OD021832. The ALS-ENABLE beamlines are supported in part by the National Institutes of Health, National Institute of General Medical Sciences, grant P30 GM124169. SAXS data collection was also conducted at the Advanced Light Source, at the SYBLS beamline supported by NIH project ALS-ENABLE (P30 GM124169) and a High-End Instrumentation Grant S10 OD018483. Results shown in this report are derived from work performed at Structural Biology Center funded by the U.S. Department of Energy, Office of Biological and Environmental Research, and operated for the DOE Office of Science at the Advanced Photon Source by Argonne National Laboratory under Contract No. DE-AC02-06CH11357.

## Author contributions

All computational design work was conducted by P.B. and R.D.K. All constructs described in this manuscript were expressed and purified by L.A.D and B.T. Crystallographic analysis of kcTRP3d.1, kcTRP3d.2 and kcTRP3d.3 were conducted by B.T. and L.A.D. Crystallographic analysis of kcTRP5b.1 was conducted by L.A.D. AFM analysis was conducted by L.F.M. Chemical denaturation analysis were performed by C.T.O., J.J., and S.E.J. SAXS was conducted by R.D.K. P.B., B.L.S., and L.A.D. wrote the initial draft of the manuscript, which was then rewritten and finalized through the joint efforts of all the coauthors.

## Competing interests

P.B. and B.L.S. are named inventors on issued and pending patents describing de novo designed circular tandem repeat proteins ('cTRPs') and receive licensing revenue from IP related to those patents. The remaining authors declare no competing interests.
