## [Peer Review File · Nature Communications]

De novo design of knotted tandem repeat proteinsREVIEWER COMMENTS

Reviewer #1 (Remarks to the Author):

In their paper “De novo design of knotted tandem repeat proteins” Doyle et al. engineer artificial proteins which contain trefoil knots. Structures were confirmed with X ray crystallography and tested for stability and folding properties. An attempt to create an even more complex pentafoil structure failed and resulted in trefoil folds.

In my opinion this paper constitutes a landmark achievement which could usher in a new era of topological protein engineering, and I strongly support its publication in Nature Communications. Potential applications of the methodology are manifold and include construction of particularly stable knotted conformations which take advantage of topology to perform specific tasks. As already indicated in the manuscript a thorough investigation of artificial protein knots could also help unravel some of the fundamental questions arising from naturally occurring protein knots such as the hypothesized resistance against thermal or proteasomal degradation and address questions arising from folding topologically challenging structures. Even though this is only the second consciously created artificial protein knot after King et al, PNAS (2010), this novel approach did not rely on naturally occurring symmetries, but designs protein knots de novo.

My only suggestion concerns the embedding of these impressive results in the context of very recent developments in the field of protein topology, e.g., with respect to knots predicted by alphafold (Brems et al, Protein Science 2022, Perlinska et al, Protein Science 2023). These references list a multitude of naturally occurring candidate structures for protein knots which could be resolved with experiments. Likewise, there is a very recent (so far unpublished) report on the successful structure determination of a composite trefoil knot which should be discussed as well (<https://doi.org/10.1101/2023.03.13.532328>).

Reviewer #2 (Remarks to the Author):

This paper describes the design and structural/biophysical characterization of novel (torus) knotted protein topologies based on repetitive helical components. The ideas are novel and the challenging, involving high levels of expertise on the design side as well as the structural/biophysical side. Among the design trials undertaken, there are some interesting successes and some insightful results and interpretations. For the most part, the experimental analyses appear sound. The paper will add substantially to the literature in the areas of protein design and complex topologies. For the crystallographic investigations, there are some nuances regarding symmetry and pseudosymmetry that need to be analyzed in more detail prior to publication. Those issues are discussed below, along with several minor suggestions for clarifying edits.

CRYSTALLOGRAPHIC ANALYSIS

a) At the end of the first paragraph on the top of page 6, more needs to be said about: whether the search model was a complete trefoil or just a unique third of the trefoil, and what was found in the

asymmetric unit in the three different trefoil crystal structures. It seems that the P43 and I222 cases have one trefoil in the ASU while the P213 crystal has just 1/3 of a trefoil in the asymmetric unit?

b) At least a couple of additional paragraphs with further crystallographic analysis are needed right after that to deal with issues of pseudo symmetry and how the correct structures (compared to very slightly different alternatives) were proven. The P43 and I222 cases present one type of challenge and the P213 case presents a different kind of challenge; it's not clear that the authors have appropriately wrestled with the challenges in either case.

For the P43 and I222 case, there must be three almost identical crystal structure solutions, related by 120 rotations that would exchange the repeats and their connectivities/termini. Initial solutions were obtained by molecular replacement (presumably with a full trefoil as the search model?). There is little chance that molecular replacement (especially at the rotation function stage) by itself would be able to discriminate the correct orientation vs the other two alternatives, since only the connecting peptide bonds between repeats 1/2 and 2/3 and the few extra residues at the true N-terminus would be different. So how did the authors discern the correct model after the fact of automatic molecular replacement? Nothing is said on this point, raising the suspicion that it might not have been examined as carefully as necessary. To look retrospectively, the authors would need to consider a few calculations. Difference maps based on the refined model might show up positive and negative peaks at certain peptide bond connections between repeats (or at the true N-terminus where there are about 6 extra [probably flexible] residues. If significant positive/negative density peaks show up then this would guide the authors to instead generate an alternate model and perform a final refinement. If significant difference density peaks do not compel an alternative model, then it would make sense to delete connecting residues between repeats, relax/jiggle and refine, and then examine a fresh set of density maps to see if they clearly establish the correct model connections. If the authors are not able to demonstrate through careful density map analysis what the right model choice is, then it might be necessary to attempt to refine the three alternatives to see if one choice behaves better than the other two in terms of R and R_{free}. If these analyses are not definitive, then it might be necessary to consider that the crystal structure needs to be modeled as a case of static disorder between three alternative structures.

For the P213 case, the nuances are different. If I understand this situation, the trefoil is sitting on a 3-fold symmetry axis, though the molecule itself cannot strictly obey 3-fold symmetry. In this case, one has to propose a case of static disorder (at least as interpreted since the data have been reduced in tetrahedral symmetry). But how have the authors refined the structure? This is important because the reported refinement R-values are considerably worse than one expects at the high resolution obtained (1.49 Ang). There is clearly something that doesn't quite add up with this structure interpretation, and the issue of pseudo symmetry is a likely culprit. Assuming there is static disorder and the trefoil sits with equal probability in three alternative orientations, there may be no way to get a great R-value in the end, but the situation should be discussed. As a technical aside, have the authors deposited this as a full trefoil with the understanding that application of the crystal symmetry would give collisions between molecules (which one would have to ignore)?

It's possible I haven't interpreted the crystallographic situations exactly right, but it's my best guess for filling in the gaps.

MINOR POINTS

- Abstract, line 6: Change to read, "protein design in order to construct", so as to avoid the possible misinterpretation that the previously described algorithms were already applied to deeply knotted proteins.
- Abstract: The penultimate sentence lacks parallel structure. Instead, change to "sequence); two of the four repeat-repeat units matched the designed backbone"
- Page 3, 9 lines from the bottom. Where ref 13 shows up, consider citing King et al. JMB 2007, which discovered the broad category of slip knot topologies and also introduced the knot matrix (later called 'knot plot') presentation for analyzing the depth and core of protein knots.
- On page 4, at the end of the first Results paragraph, consider adding: "Note that p and q must be relatively prime in order for the resulting structure to be comprised of a single protein chain with rotational symmetry of order q."
- At the end of the first paragraph on page 5, add a sentence to give the MW of the designed proteins.
- 3 lines from the bottom of page 7, change "which induced" to "to induce".
- Last sentence on page 7. Rephrase.
- page 9, line 6: set off "as there are no prolines in kcTRP3d.3" by commas.
- Last sentence of paragraph 2 on page 11, consider changing ", however," to "; but", with no comma after 'but'
- 9 lines from the bottom of page 11: This suggests that "certain topologies" might be more problematic. It would be less vague here to make the obvious point that more complex knots are likely going to be harder, since folding pathways are going to require more gymnastics. Consider adding ", e.g. for knots with a crossing number greater than three" to the end of that sentence.
- On page 12, paragraphs 3 and 4: change "what" to "that".
- Page 14, lines 9-11: Rephrase this sentence.
- The connectivity of the pentafoil structure in Figure 6 could be made clearer. The ribbon structures in panel c could be made smaller to leave space for enlarging or redrawing the arrow insets below. From a pure geometry perspective, I think it might be clearer to draw a regular pentagon and then draw the intended star-shaped path (1 -> 3 -> 5 -> 2 -> 4) on the vertices for the left model and the alternate 1 -> 3 -> 4 -> 5 -> 2 vertex path for the observed structure.

Reviewer #3 (Remarks to the Author):

As I have expertise in AFM I was asked by the editor to review the AFM sections of the manuscript. The rest is beyond the scope of my expertise so I leave that to other referees. I read all the AFM parts, including figure 4 and caption, Mat&Meth, and Results section. I could not find anything about AFM experiments in the Discussion section. As far as my knowledge goes, all the authors wrote about AFM is technically correct, the figure is clear, the methods well described, and the results make sense.

Two more specific comments:

- There is a statement at the bottom of page 6: "The tensile strength of this construct is considerably higher than other known purely alpha-helical domains such as spectrin (around 40-50 pN)". According to the data it is higher indeed (50-100 pN). I just wonder why, is it the knot that is responsible for that higher unfolding force? If yes, how?
- There is zero discussion about AFM experiments, so I am not sure I understand why they have been performed. Is it just to demonstrate the presence of the knot in the expressed construct? If yes, perhaps that should be explicitly stated somewhere in the text.

Response to reviewer comments.

We greatly appreciate the time and effort taken by the editor and reviewers to evaluate our manuscript. In response to their comments, questions, and suggestions, we have made multiple changes to the text, as detailed below inline with the comments (reviewer comments shown in italics).

REVIEWER COMMENTS

Reviewer #1 (Remarks to the Author):

In their paper "De novo design of knotted tandem repeat proteins" Doyle et al. engineer artificial proteins which contain trefoil knots. Structures were confirmed with X ray crystallography and tested for stability and folding properties. An attempt to create an even more complex pentafoil structure failed and resulted in trefoil folds. In my opinion this paper constitutes a landmark achievement which could usher in a new era of topological protein engineering, and I strongly support its publication in Nature Communications. Potential applications of the methodology are manifold and include construction of particularly stable knotted conformations which take advantage of topology to perform specific tasks. As already indicated in the manuscript a thorough investigation of artificial protein knots could also help unravel some of the fundamental questions arising from naturally occurring protein knots such as the hypothesized resistance against thermal or proteasomal degradation and address questions arising from folding topologically challenging structures. Even though this is only the second consciously created artificial protein knot after King et al, PNAS (2010), this novel approach did not rely on naturally occurring symmetries, but designs protein knots de novo.

We appreciate the reviewer's positive comments on the manuscript.

My only suggestion concerns the embedding of these impressive results in the context of very recent developments in the field of protein topology, e.g., with respect to knots predicted by alphafold (Brems et al, Protein Science 2022, Perlinska et al, Protein Science 2023). These references list a multitude of naturally occurring candidate structures for protein knots which could be resolved with experiments. Likewise, there is a very recent (so far unpublished) report on the successful structure determination of a composite trefoil knot which should be discussed as well (<https://doi.org/10.1101/2023.03.13.532328>).

These are excellent suggestions. We have added additional references and text to the Discussion section (first and third paragraphs).

Reviewer #2 (Remarks to the Author):

This paper describes the design and structural/biophysical characterization of novel (torus) knotted protein topologies based on repetitive helical components. The ideas are novel and the challenging, involving high levels of expertise on the design side as well as the structural/biophysical side. Among the design trials undertaken, there are some interesting successes and some insightful results and interpretations. For the most part, the experimental analyses appear sound. The paper will add substantially to the literature in the areas of protein design and complex topologies. For the crystallographic investigations, there are some nuances regarding symmetry and pseudosymmetry that need to be analyzed in more detail prior to publication. Those issues are discussed below, along with several minor suggestions for clarifying edits.

We appreciate the reviewer's positive comments on the manuscript. We have added additional details on the crystallographic analyses to the "Knotted trefoil designs and characterization" sub-section of the Results. We agree that these are subtle issues that could benefit from greater clarity. We are also sensitive to the length

and complexity of the manuscript which contains multiple computational and experimental approaches. We are confident that the crystallographic analyses as now presented support the existence of the trefoil-knotted topology in the designed proteins, which we view as the main point of those studies.

CRYSTALLOGRAPHIC ANALYSIS

a) At the end of the first paragraph on the top of page 6, more needs to be said about: whether the search model was a complete trefoil or just a unique third of the trefoil, and what was found in the asymmetric unit in the three different trefoil crystal structures. It seems that the P43 and I222 cases have one trefoil in the ASU while the P213 crystal has just 1/3 of a trefoil in the asymmetric unit?

The reviewer is correct that kcTRP3d.1 (P 43) and kcTRP3d.3 (I 2 2 2) contain one trefoil in the ASU while kcTRP3d.2 (P 21 3) contains 1/3 of the trefoil, equivalent to a single helix-turn-helix repeat. We have clarified this in the manuscript and added additional information about the search model used for each design.

b) At least a couple of additional paragraphs with further crystallographic analysis are needed right after that to deal with issues of pseudo symmetry and how the correct structures (compared to very slightly different alternatives) were proven. The P43 and I222 cases present one type of challenge and the P213 case presents a different kind of challenge; it's not clear that the authors have appropriately wrestled with the challenges in either case.

We appreciate the reviewer's comments and attention to crystallographic detail, as well as the discussion of the nuances in these structures.

For the P43 and I222 case, there must be three almost identical crystal structure solutions, related by 120 rotations that would exchange the repeats and their connectivities/termini. Initial solutions were obtained by molecular replacement (presumably with a full trefoil as the search model?). There is little chance that molecular replacement (especially at the rotation function stage) by itself would be able to discriminate the correct orientation vs the other two alternatives, since only the connecting peptide bonds between repeats 1/2 and 2/3 and the few extra residues at the true N-terminus would be different. So how did the authors discern the correct model after the fact of automatic molecular replacement? Nothing is said on this point, raising the suspicion that it might not have been examined as carefully as necessary. To look retrospectively, the authors would need to consider a few calculations. Difference maps based on the refined model might show up positive and negative peaks at certain peptide bond connections between repeats (or at the true N-terminus where there are about 6 extra [probably flexible] residues. If significant positive/negative density peaks show up then this would guide the authors to instead generate an alternate model and perform a final refinement. If significant difference density peaks do not compel an alternative model, then it would make sense to delete connecting residues between repeats, relax/jiggle and refine, and then examine a fresh set of density maps to see if they clearly establish the correct model connections. If the authors are not able to demonstrate through careful density map analysis what the right model choice is, then it might be necessary to attempt to refine the three alternatives to see if one choice behaves better than the other two in terms of R and Rfree. If these analyses are not definitive, then it might be necessary to consider that the crystal structure needs to be modeled as a case of static disorder between three alternative structures.

Surprisingly, the structure of kcTRP3d.1 (P 43) unambiguously showed the connectivity/termini of the structure; additionally, when asked to output 3 molecular replacement solutions Phaser output only two with one having a higher log-likelihood gain and translation Z-score. Nonetheless, residues between repeating units were deleted and refinement performed with simulated annealing to remove model bias. The difference maps indicated that the original solution was in the correct orientation. Furthermore, towards the end of the refinement process,

minor density at the N-terminal corresponding to 2-3 additional residues was present, albeit not at a level to allow accurate model building.

The reviewer is correct that there is ambiguity between three alternative structures related by 120 degree rotations for kcTRP3d.3 (1 2 2 2). We have examined this ambiguity as part of the structure solution process (closely mirroring the reviewer's suggestions) but felt that a deep dive into this was outside the scope of the paper as we were ultimately asking if the structures display the intended knotted topologies, to which the answer would not change if the structure was rotated by plus/minus 120 degrees. We have added text to the manuscript describing this ambiguity.

For the P213 case, the nuances are different. If I understand this situation, the trefoil is sitting on a 3-fold symmetry axis, though the molecule itself cannot strictly obey 3-fold symmetry. In this case, one has to propose a case of static disorder (at least as interpreted since the data have been reduced in tetrahedral symmetry). But how have the authors refined the structure? This is important because the reported refinement R-values are considerably worse than one expects at the high resolution obtained (1.49 Ang). There is clearly something that doesn't quite add up with this structure interpretation, and the issue of pseudo symmetry is a likely culprit. Assuming there is static disorder and the trefoil sits with equal probability in three alternative orientations, there may be no way to get a great R-value in the end, but the situation should be discussed. As a technical aside, have the authors deposited this as a full trefoil with the understanding that application of the crystal symmetry would give collisions between molecules (which one would have to ignore)?

The reviewer is correct that kcTRP3d.2 (P 21 3) sits on the 3-fold crystallographic axis, representing an average of the repeating subunits as opposed to a designated individual repeating subunit (i.e. helix-turn-helix subunit 1). We have deposited the structure with the biological assembly equivalent to a full trefoil, with the understanding that one needs to apply crystallographic symmetry to achieve the full molecule. We agree that the pseudo-symmetry is to blame for the higher-than-average R-values. In addition to the three repeating subunits being averaged into a single representation, several N-terminal residues (artifacts of affinity tag removal) were also averaged out. During data processing and structure solution we attempted to process and refine the data into a lower order space group to determine if the R-values, and other metrics, would improve when the entire trefoil was contained within the ASU. Ultimately, it was found that the P 21 3 space group produced the best overall structure in regard to both statistics and electron density.

It's possible I haven't interpreted the crystallographic situations exactly right, but it's my best guess for filling in the gaps.

We greatly appreciate the reviewer's effort and attention to detail here, and we apologize for the lack of clarity in the original submission.

MINOR POINTS

- Abstract, line 6: Change to read, "protein design in order to construct", so as to avoid the possible misinterpretation that the previously described algorithms were already applied to deeply knotted proteins.

Thanks, this has been done.

- Abstract: The penultimate sentence lacks parallel structure. Instead, change to "sequence); two of the four repeat-repeat units matched the designed backbone"

Done.

- Page 3, 9 lines from the bottom. Where ref 13 shows up, consider citing King et al. JMB 2007, which discovered the broad category of slip knot topologies and also introduced the knot matrix (later called 'knot plot') presentation for analyzing the depth and core of protein knots.

Done.

- On page 4, at the end of the first Results paragraph, consider adding: "Note that p and q must be relatively prime in order for the resulting structure to be comprised of a single protein chain with rotational symmetry of order q ."

Done.

- At the end of the first paragraph on page 5, add a sentence to give the MW of the designed proteins.

Done.

- 3 lines from the bottom of page 7, change "which induced" to "to induce".

Done.

- Last sentence on page 7. Rephrase.

Done.

- page 9, line 6: set off "as there are no prolines in kcTRP3d.3" by commas.

Done.

- Last sentence of paragraph 2 on page 11, consider changing ", however," to ", but", with no comma after 'but'

Done.

- 9 lines from the bottom of page 11: This suggests that "certain topologies" might be more problematic. It would be less vague here to make the obvious point that more complex knots are likely going to be harder, since folding pathways are going to require more gymnastics. Consider adding ", e.g. for knots with a crossing number greater than three" to the end of that sentence.

We agree and have added further text and references here.

- On page 12, paragraphs 3 and 4: change "what" to "that".

Done, assuming the reviewer meant "which" to "that".

- Page 14, lines 9-11: Rephrase this sentence.

Done.

- The connectivity of the pentafoil structure in Figure 6 could be made clearer. The ribbon structures in panel c could be made smaller to leave space for enlarging or redrawing the arrow insets below. From a pure geometry perspective, I think it might be clearer to draw a regular pentagon and then draw the intended star-shaped path (1 -> 3 -> 5 -> 2 -> 4) on the vertices for the left model and the alternate 1 -> 3 -> 4 -> 5 -> 2 vertex path for the observed structure.

We experimented with this suggestion but could not arrive at an improved visualization, in part because shrinking the ribbon structures also shrinks the colored arrows in their centers, which makes it harder to recognize the link between them and the colored arrow diagrams below. So, in the end, we decided to maintain the current figure.

Reviewer #3 (Remarks to the Author):

As I have expertise in AFM I was asked by the editor to review the AFM sections of the manuscript. The rest is beyond the scope of my expertise so I leave that to other referees. I read all the AFM parts, including figure 4 and caption, Mat&Meth, and Results section. I could not find anything about AFM experiments in the Discussion section. As far as my knowledge goes, all the authors wrote about AFM is technically correct, the figure is clear, the methods well described, and the results make sense.

We appreciate the time and effort taken by the reviewer to evaluate our manuscript.

Two more specific comments:

- There is a statement at the bottom of page 6: "The tensile strength of this construct is considerably higher than other known purely alpha-helical domains such as spectrin (around 40-50 pN)". According to the data it is higher indeed (50-100 pN). I just wonder why, is it the knot that is responsible for that higher unfolding force? If yes, how?

We have added additional discussion of this point ("Single molecule unfolding studies" in the Results); our current thinking is that it may be other aspects of the computational design process that are imparting the enhanced tensile strength, though we cannot be sure.

- There is zero discussion about AFM experiments, so I am not sure I understand why they have been performed. Is it just to demonstrate the presence of the knot in the expressed construct? If yes, perhaps that should be explicitly stated somewhere in the text.

The AFM experiments were performed in order to further characterize this de novo designed knotted protein.

REVIEWERS' COMMENTS

Reviewer #2 (Remarks to the Author):

The authors have revised the manuscript adequately and it is now suitable for publication.

Reviewer #3 (Remarks to the Author):

To my comment:

"There is zero discussion about AFM experiments, so I am not sure I understand why they have been performed. Is it just to demonstrate the presence of the knot in the expressed construct? If yes, perhaps that should be explicitly stated somewhere in the text."

The authors made this very short response:

"The AFM experiments were performed in order to further characterize this de novo designed knotted protein."

Ok but what does this characterization has to offer to our understanding of this knotted protein? Not clear to me... Although I understand the AFM community is not the primary target of this paper, doesn't it deserve a couple of lines somewhere in the manuscript?

Response to reviewer comments

We appreciate the time and effort taken by the editor and reviewers to assess the revised version of our manuscript on knotted protein designs. In response to their comments, we have added further discussion to contextualize the AFM results, as detailed below.

Reviewer #3 (Remarks to the Author):

To my comment:

"There is zero discussion about AFM experiments, so I am not sure I understand why they have been performed. Is it just to demonstrate the presence of the knot in the expressed construct? If yes, perhaps that should be explicitly stated somewhere in the text."

The authors made this very short response:

"The AFM experiments were performed in order to further characterize this de novo designed knotted protein."

Ok but what does this characterization has to offer to our understanding of this knotted protein? Not clear to me... Although I understand the AFM community is not the primary target of this paper, doesn't it deserve a couple of lines somewhere in the manuscript?

We agree and apologize for not giving this experiment sufficient attention in the original manuscript. We have added additional text addressing and interpreting the AFM results to the discussion, as indicated in red below.

"Similar topologies in naturally evolved (and previously engineered) proteins. A comprehensive analysis of the structure, stability, **mechanical unfolding and knot tightening, under force**, unfolding and refolding kinetics of the designed trefoil knotted proteins described here, enables a comparative analysis of the structure and properties of these proteins with naturally occurring (and several previously engineered constructs, described above) left- and right-handed trefoil-knotted proteins in the KnotProt database".

"The superstability of kcTRP3d.3 is also reflected in the half-life of unfolding in water, which is estimated to be some 100 years – far greater than measured for any other knotted protein." It is interesting to note that despite the considerable thermodynamic stability measured with chemical denaturants, the mechanical stability of kcTRP3d.3 measured by AFM is similar to those observed for other knotted proteins, as well as within the range of forces measured for unknotted proteins [Ref 16]. However, the tightened knot in kcTRP3d.3 is somewhat larger than those observed previously for either a 4₁-knotted phytochrome [New ref: Bornschlogl paper], or a 3₁-knotted unfolded state of UCH-L1, and in fact is closer in size to the 5₂-knotted unfolded state of UCH-L1 [New ref: Ziegler paper]. It is likely that this is because of the exact location of the knot in the unfolded state and the size of the side chains within the tightened knot."

New references:

Tightening the knot in phytochrome by single-molecule atomic force microscopy.

Bornschlöggl T, Anstrom DM, Mey E, Dzubiella J, Rief M, Forest KT. Biophys J. 2009 Feb 18;96(4):1508-14.

Knotting and unknotting of a protein in single molecule experiments.

Ziegler F, Lim NC, Mandal SS, Pelz B, Ng WP, Schlierf M, Jackson SE, Rief M. Proc Natl Acad Sci U S A. 2016 Jul 5;113(27):7533-8.